# Cryo-EM structure and evolutionary history of the conjugation surface exclusion protein TraT

Chloe Seddon[1,2,5], Sophia David [3,5], Joshua L. C. Wong [1], Naito Ishimoto [1,2], Shan He [1], Jonathan Bradshaw[1], Wen Wen Low[1,4], Gad Frankel [1] ✉ & Konstantinos Beis [1,2] ✉

Conjugation plays a major role in dissemination of antimicrobial resistance genes. Following transfer of IncF-like plasmids, recipients become refractory to a second wave of conjugation with the same plasmid via entry (TraS) and surface (TraT) exclusion mechanisms. Here, we show that TraT from the pKpQIL and F plasmids (TraT$_{pKpQIL}$ and TraT$_F$) exhibits plasmid surface exclusion specificity. The cryo-EM structures of TraT$_{pKpQIL}$ and TraT$_F$ reveal that they oligomerise into decameric champagne bottle cork-like structures, which are anchored to the outer membrane via a diacylglycerol and palmitic acid modified α-helical barrel domain. Unexpectedly, we identify chromosomal TraT homologues from multiple Gram-negative phyla which form numerous divergent lineages in a phylogenetic tree of TraT sequences. Plasmid-associated TraT sequences are found in multiple distinct lineages, including two separate clades incorporating TraT from Enterobacteriaceae IncF/F-like and Legionellaceae F-like plasmids. These findings suggest that different plasmid backbones have acquired and co-opted TraT on independent occasions.

Bacterial conjugation is a form of horizontal gene transfer that describes contact-dependent unidirectional transfer of self-transmissible plasmids, mobilisable plasmid and Hfr (high frequency recombinant) from donor to recipient bacteria[1,2]. Conjugation can occur in any environment (e.g. gut, plant surfaces, agricultural settings, soil) in both Gram negative and Gram positive bacteria[3]. DNA transfer is mediated by a large nanomachine embedded within the donor cell envelope, the type IV secretion system (T4SS)[4,5], which is extended by a hollow, also known as the sex, pilus[6,7]. The T4SSs are encoded by conjugative plasmids across different incompatibility (Inc) groups[8].

IncF plasmids, found among Enterobacterales, often encode virulence determinants and antibiotic resistance genes[9]. They include the *Salmonella enterica* plasmid pSLT[10], encoding type III secretion system (T3SS) effectors, the enteropathogenic *Escherichia coli* (EPEC) plasmid pMAR7[11], encoding the bundle-forming pilus, the contemporary *K. pneumoniae* plasmid pKpQIL, encoding carbapenem resistance[12] and the classical *E. coli* F plasmid[1]. In the prevailing model of IncF plasmid conjugation, the extended and flexible conjugative pilus on the donor contacts a recipient in its proximity[13], a process known as mating pair formation (MPF). Upon formation of MPF, which could mediate inefficient plasmid transfer[14,15], the pilus retracts back towards the donor, enabling the donor and recipient to form a tight 'mating junction' at their membrane interfaces, a process termed mating pair stabilisation (MPS)[16,17]. We have recently shown that MPS is mediated through interactions between TraN, a plasmid-encoded

[1]Department of Life Sciences, Imperial College London, London SW7 2AZ, UK. [2]Rutherford Appleton Laboratory, Research Complex at Harwell, Didcot, Oxfordshire OX11 0FA, UK. [3]Centre for Genomic Pathogen Surveillance, Pandemic Sciences Institute, University of Oxford, Oxford OX3 7DQ, UK. [4]Present address: Department of Biochemistry, Yong Loo Lin School of Medicine, National University of Singapore, Singapore, Singapore. [5]These authors contributed equally: Chloe Seddon, Sophia David. ✉e-mail: g.frankel@imperial.ac.uk; kbeis@imperial.ac.uk

outer membrane protein (OMP) in the donor, of which there are at least four isotypes (α, β, γ, δ) and an OMP in the recipient. Conjugation species specificity and host range is mediated by specific pairings such as TraNα-OmpW, TraNβ-OmpK36, TraNγ-OmpA and TraNδ-OmpF[3,17].

Following plasmid transfer via the T4SS in the donor and an unknown DNA conduit in the recipient, the latter is no longer able to serve as an efficient recipient for subsequent rounds of conjugation for the same plasmid[18]. In IncF plasmids this process is mediated by entry and surface exclusion (EEX and SFX respectively) and is crucial for preventing lethal zygosis (LZ) that is attributable to repeated rounds of conjugation[19]. EEX, which is exhibited by multiple conjugative plasmids of both Gram-positive and Gram-negative bacteria, is mediated in IncF plasmids by the inner membrane protein TraS[20]. SFX, which is mediated by the OMP TraT, is specific for Gram-negative bacteria[21]. In terms of efficiency, EEX plays a more prominent role in exclusion, while TraT-mediated SFX of the F plasmid was reported to reduce plasmid entry by 10–50 fold[13].

TraT is a highly expressed -24 kDa lipoprotein with a copy number of approximately 29,000–84,000 copies/cell[13,22]. It is post-translationally modified by the covalent attachment of diacylglycerol (DAG) and palmitic acid (PA) molecules to the sulfhydryl group of the primary cysteine residue of the mature protein[13]. In addition to SFX, TraT has been implicated in disaggregating mating pairs after DNA transfer[16] and in serum resistance[23]. Although TraT has been identified in plasmids of Gram negative bacterias, there are no reports to show its existence outside this context.

Multiple SFX models have been proposed, including interference with MPF or MPS. However, TraT-mediated SFX was shown to be unaffected by the pilin or the TraN isoform expressed in the donor[24,25].

Recipients ectopically expressing TraT are capable of SFX, which could also be mediated by adding purified TraT to mating mixtures[21]. However, the molecular basis of SFX remains elusive.

In this work, we determined the cryo-EM structure of TraT encoded by pKpQIL (TraT$_{pKpQIL}$) and the F (TraT$_F$) plasmid derivative, pOX38, at 2.47 Å and 2.66 Å resolution respectively. This revealed that lipidated TraT oligomerises into a decameric champagne cork-like structure, composing of a transmembrane α-helical barrel domain and an extracellular ring domain. Unexpectedly, we also identified TraT homologues across multiple Gram-negative phyla, including within the chromosomes of diverse species, and show that the *traT* genes in *Enterobacteriaceae* IncF/F-like and *Legionella* spp. F-like plasmids have independent origins.

## Results
### TraT-mediated SFX is plasmid specific
To establish the degree of SFX by TraT$_{pKpQIL}$, we quantified conjugation efficiency of pKpQIL from *K. pneumoniae* strain ICC8001 donor into an ICC8001 recipient containing pBAD-*traT$_{pKpQIL}$* or an empty pBAD vector as a control. The conjugation frequency of pKpQIL into recipients expressing TraT$_{pKpQIL}$ was reduced by a log-fold, −3.6, compared to the baseline level of conjugation, −2.5, in the absence of TraT$_{pKpQIL}$ (Fig. 1a). We next investigated the specificity of TraT by measuring pKpQIL conjugation frequency into ICC8001 recipients containing pBAD-*traT$_F$*. The conjugation frequency of pKpQIL into ICC8001-pBAD-*traT$_F$* and ICC8001-pBAD-empty recipients did not display significant differences, suggesting that TraT$_F$ cannot exclude pKpQIL (Fig. 1b). Our results further support previous findings that SFX is plasmid specific[26].

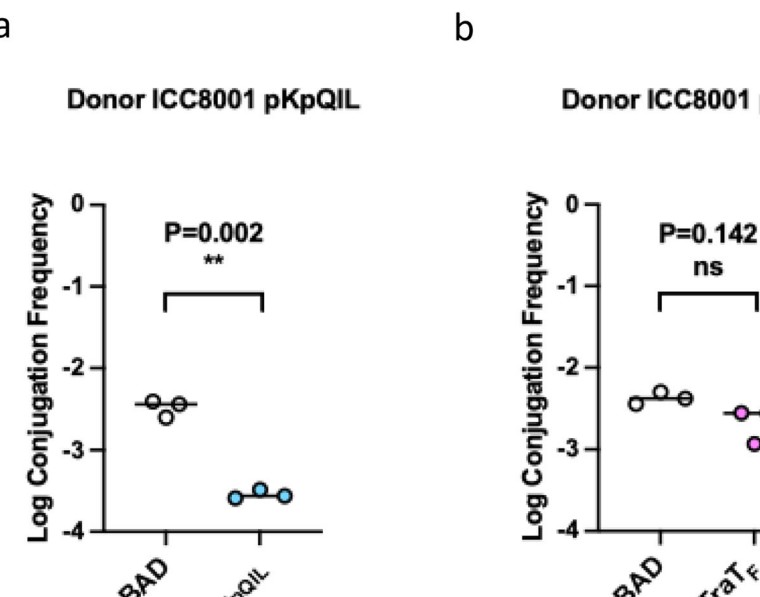

**Fig. 1 | SFX is species specific. a** The effect of TraT$_{pKpQIL}$-mediated pKpGFP SFX. The log conjugation frequency of the pKpQIL plasmid into ICC8001 recipients carrying either pBAD or pBAD-TraT$_{pKpQIL}$ vector. The presence of TraT$_{pKpQIL}$ in the recipient produces a log-fold reduction in pKpGFP conjugation. **b** The effect of TraT$_F$ on pKpQIL SFX. The log conjugation frequency of the pKpQIL plasmid into ICC8001 recipients carrying either pBAD or pBAD-TraT$_F$ are shown. The presence of TraT$_F$ on the recipient does not affect the log conjugation frequency of pKpQIL. Data were statistically analysed by a two paired *t*-test. Statistical significance is marked by ** (*p* = 0.002) and ns (*p* = 0.142) indicates non significance. Data presented is representative of three biological repeats with the individual repeats and the overall average for each data set being shown. Source data are provided as a Source Data file.

## TraT$_{pKpQIL}$ forms a decameric outer membrane complex

To gain insights into the molecular mechanism of SFX, we purified recombinant TraT$_{pKpQIL}$ from *E. coli* OM vesicles in CYMAL-6 and determined the cryo-EM structure. Analysis of the purified TraT$_{pKpQIL}$ by mass spectrometry confirmed that the first cysteine of the mature protein (C36) is modified by DAG and PA, which agrees with previous reports (Supplementary Fig. 1)[22]. The TOF MS ES+ analysis showed a peak corresponding to a molecular weight of 25,580. The molecular weight of mature TraT$_{pKpQIL}$ is 24,749.87 Da (23,735.81 Da TraT$_{pKpQIL}$ and 1014.06 Da of residues from cloning); the unaccounted 830.13 Da mass can be attributed to the posttranslational modification by DAG and PA with an average molecular weight of around 300–700 Da and 256 Da, respectively.

We determined the cryo-EM structure of TraT$_{pKpQIL}$ at an overall resolution of 2.47 Å (Fourier shell correlation (FSC) = 0.143 criterion; Fig. 2a, Supplementary Fig. 2 and Supplementary Table 1). The map displays 10-fold symmetry corresponding to ten copies of TraT$_{pKpQIL}$. We have built an atomic model for the oligomeric TraT$_{pKpQIL}$ that is composed of a transmembrane α-helical barrel domain and an extracellular ring-like domain; the overall structure resembles a champagne bottle cork (Fig. 2b). A belt of featureless density surrounds the exterior and the interior of the α-helical barrel domain that represents the detergent micelle and defines the transmembrane region of TraT; the barrel is shielded, and it does not form a pore in the OM. The role of the transmembrane α-helical barrel domain is to anchor TraT to the OM. Each TraT$_{pKpQIL}$ protomer consists of three amphipathic TM helices (α1, α3 and α4) that anchors it to the OM, a central β-sandwich domain (β1–β7) that is flanked by two α-helices (α2 and α5) that protrudes to the extracellular space; β5 and β6 form a β-hairpin perpendicular to the top of the β-sandwich domain (Fig. 2c). We also observe density for the glycerol backbone of DAG, and partial density for the two acyl chains; no density could be observed for the PA (Fig. 2d). Each protomer associates with one DAG molecule, where interactions occur exclusively between DAG and its own protomer. DAG is found in the interface of the TM helices α1, α3 and α4 (Fig. 2d) and forms a hydrogen bond between its diacylglycerol moiety and the backbone of T137. Similarly, Van der Waals interactions are formed between A38, G167 and N236 and the acyl chains of DAG.

The decameric structure is predominantly stabilised by interactions between the extracellular domain of the protomers; the α-helical barrel domain does not display significant interactions with adjacent protomers. The amphipathic α-helical region is stabilised by hydrogen bonds between α1 from one protomer and α4 of the neighbouring one (Fig. 3). The β-sandwich domain of each protomer is positioned adjacent to the corresponding domain of adjacent ones. In the decameric structure, the β-sandwich domain forms hydrogen bonds between α5 and β1 from one protomer and β2-4 and β7 of the adjacent β-sandwich domain (Fig. 3). The β-hairpin motif, β5 and β6, displays domain intertwining with the β-sandwich domain of the next protomer (Fig. 3). The β-hairpin motif is further stabilised by hydrogen bonds between β6 of one protomer and β5 of the adjacent protomer (Fig. 3). The arrangement between the β-sandwich domains, including the β5 and β6 intertwining, results in the formation of a cyclic architecture that consists of an inner β-barrel with a diameter of 48 Å (Fig. 2b). The interior of the β-barrel is lined by a belt of negatively and positively charged residues (Supplementary Fig. 3).

We determined the importance of the DAG and PA modifications and the role of the TM helix α1 for TraT oligomerisation and insertion to the OM by designing a mutant construct C36S (TraT$_{C36S}$) that cannot be modified by DAG/PA, a construct lacking both C36 and α1 (TraT$_{ΔC36/α1}$) and a construct only lacking α1 (TraT$_{Δα1}$). Removal of the DAG/PA modified C36 resulted in loss of TraT expression and SFX (Fig. 4a) that could be attributed to either protein instability or incorrect processing (Fig. 4a); DAG and PA modifications are essential for the correct targeting of lipoproteins to the OM as the lipids drive insertion into the OM[27]. Retaining the modified C36 but deletion of the TM helix α1 in TraT$_{Δα1}$ resulted in the expression of TraT in the OM, but recipients expressing TraT$_{Δα1}$ confer no SFX (Fig. 4b). The lack of SFX is either due to the incorrect insertion to the OM or compromised stability relative to the wild type protein. In contrary, loss of both DAG/PA and α1 resulted in TraT$_{ΔC36/α1}$ to be found in the soluble fraction that has lost its ability to oligomerise and behaves like a monomeric protein that is migrating at around 10 ml on a Superdex S75 10/300 column, corresponding to a molecular weight of around 28 kDa (the MW of TraT$_{pKpQIL}$ is 25 kDa), while full length TraT$_{pKpQIL}$ migrates at around 10 ml on a Superdex S200 10/300 column, corresponding to an oligomer of around 300 kDa (250 kDa contributed by the TraT$_{pKpQIL}$ decamer and 50 kDa by the CYMAL-6 micelle) (Fig. 4c, d). The monomeric TraT$_{ΔC36/α1}$ is likely due to the absence of DAG and α1$_{K23}$ intercalation with α4$_{A172}$ on an adjacent protomer, leading to the loss of interactions between DAG and the interface of α3 and α4 (Fig. 2d). Deletion of α1 is likely to also destabilise α3 and α4 due to the loss of stabilising hydrogen bonds mediated by K23 (Fig. 3).

## High structural conservation between TraT$_{pKpQIL}$ and TraT$_F$

To study the structural basis of SFX specificity, the structure of TraT$_F$ was also determined. The cryo-EM structure has an overall resolution of 2.66 Å (Fourier shell correlation (FSC) = 0.143 criterion; Fig. 5a, Supplementary Fig. 4 and Supplementary Table 1). Like TraT$_{pKpQIL}$, TraT$_F$ consists of ten identical protomers that form a transmembrane α-helical barrel domain and an extracellular ring-like domain (Fig. 5b). Density corresponding to DAG was weak and it was not included in the model building (mass spectrometry analysis confirmed that recombinant mature TraT$_F$ is also modified by DAG and PA (Supplementary Fig. 1). The TraT$_F$ monomer comprises the same secondary structure elements as shown for TraT$_{pKpQIL}$ and exhibits a similar conformation (Fig. 5c). The arrangement of the TraT$_F$ protomers in the oligomer mirrors the arrangement of the TraT$_{pKpQIL}$ protomers in the TraT$_{pKpQIL}$ decamer.

While TraT$_{pKpQIL}$ and TraT$_F$ (85% amino acid identity) exhibit SFX plasmid specificity, their structures are highly conserved. TraT$_{pKpQIL}$ and TraT$_F$ can be superimposed with an rmsd of 0.523 Å over 225 Cα atoms (Fig. 5c). Most sequence differences between the two proteins are found within the extracellular region of TraT, in the β-strands lining the TraT ring opening (β2, β5 and β6) and in the flanking α-helices (α2 and α5); these differences have been mapped onto the TraT$_{pKpQIL}$ structure (Fig. 5d and Supplementary Fig. 5).

## Identification of sequence homologues reveals diverse chromosomally-encoded TraT

We next investigated the taxonomic distribution of TraT by extracting homologues from UniprotKB with a length of 200–300 amino acids, ≥30% amino acid similarity to TraT$_{pKpQIL}$ and a defined chromosomal or plasmid origin. Of 399 TraT homologues identified, 295 (73.9%) were from plasmids while the other 104 (26.1%) were identified from chromosomal sequences. Almost all the plasmid-encoded TraT sequences were from the Pseudomonadota phylum (98.0%; 289/295), with the remaining few sequences from either the Campylobacterota phylum (0.3%; 1/295) or unidentified organisms (1.7%; 5/295). Within the Pseudomonadota, 92.4% (267/289) of the sequences were from different genera in the Enterobacteriaceae family while a further 4.8% (14/289) were from the *Legionella* or *Fluoribacter* genera of the family Legionellaceae. Chromosomally-encoded TraT sequences were more widely distributed, found among seven different Gram-negative phyla. However, over half (57.7%; 60/104) were recovered from various families of the Pseudomonadota phylum, while a further 25.0% (26/104) were from the *Fusobacterium* genus of the Fusobacteriota phylum.

We constructed a maximum-likelihood phylogenetic tree of the 399 TraT protein sequences using the best-fitting evolutionary

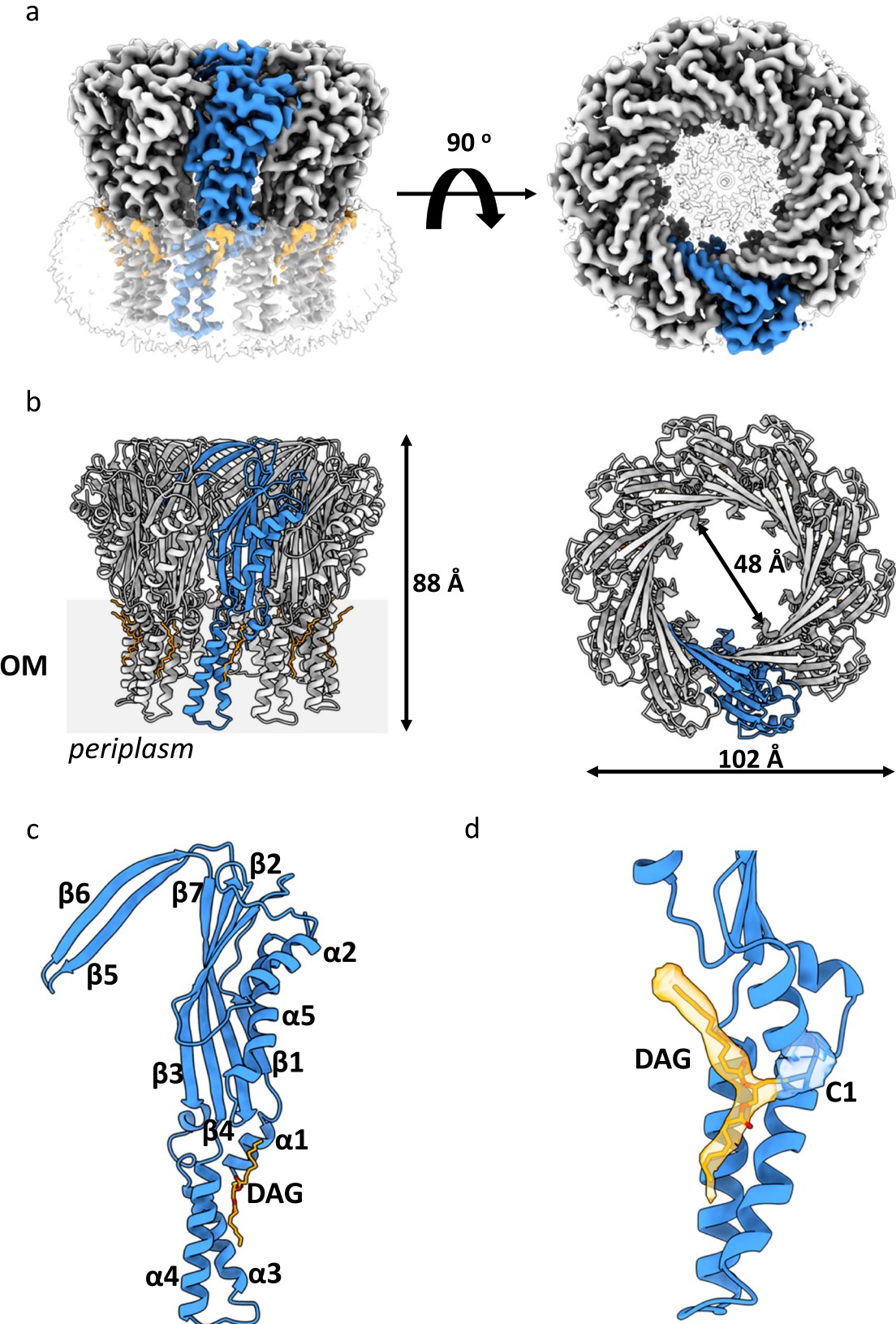

**Fig. 2 | Cryo-EM structure of TraT_pKpQIL. a** Ab initio cryo-EM map of TraT_pKpQIL at 2.47 Å resolution. The ten-fold symmetry results in a cork like structure; the TraT_pKpQIL protomers are shown in grey and one in blue. The DGA lipids and CYMAL-6 micelle are shown in orange and transparent grey, respectively. **b** Cartoon representation of the TraT_pKpQIL oligomer. Each protomer is coloured as in (**a**). The decamer consists of an α-helical barrel embedded in the OM and an extracellular domain that consists of an inner β-barrel domain. **c** The TraT_pKpQIL protomer consists of three TM helices, α1, α3 and α4, a β-sandwich domain flanked by α-helices and a β-hairpin motif. The DAG molecule is found in the interface of the TM helices. **d** Density for the C36 modified by DAG. DAG is shown as sticks; carbon atoms are shown in orange and oxygen in red.

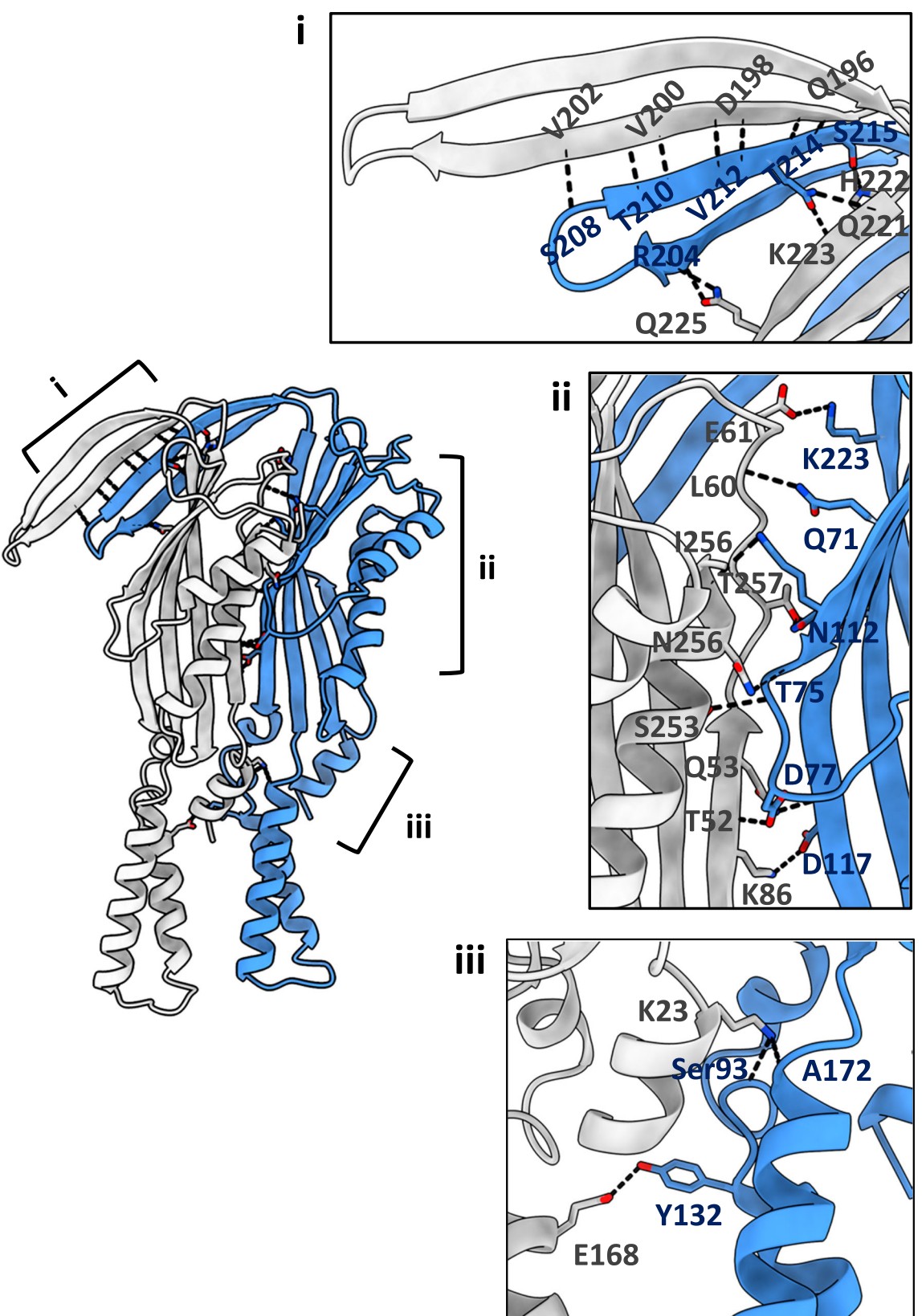

**Fig. 3 | Domain interface of TraT protomers.** The interface of two protomers is mostly stabilised by hydrogens bonds between (i) the β-hairpin motif and (ii) the β-sandwich domain. (iii) The TM α-helices of adjacent protomers are stabilised by intermolecular interactions without the contribution of interactions with DAG. The three panels show the detailed interactions along the interface of two protomers. Absence of a side chains indicate interactions with the peptide backbone.

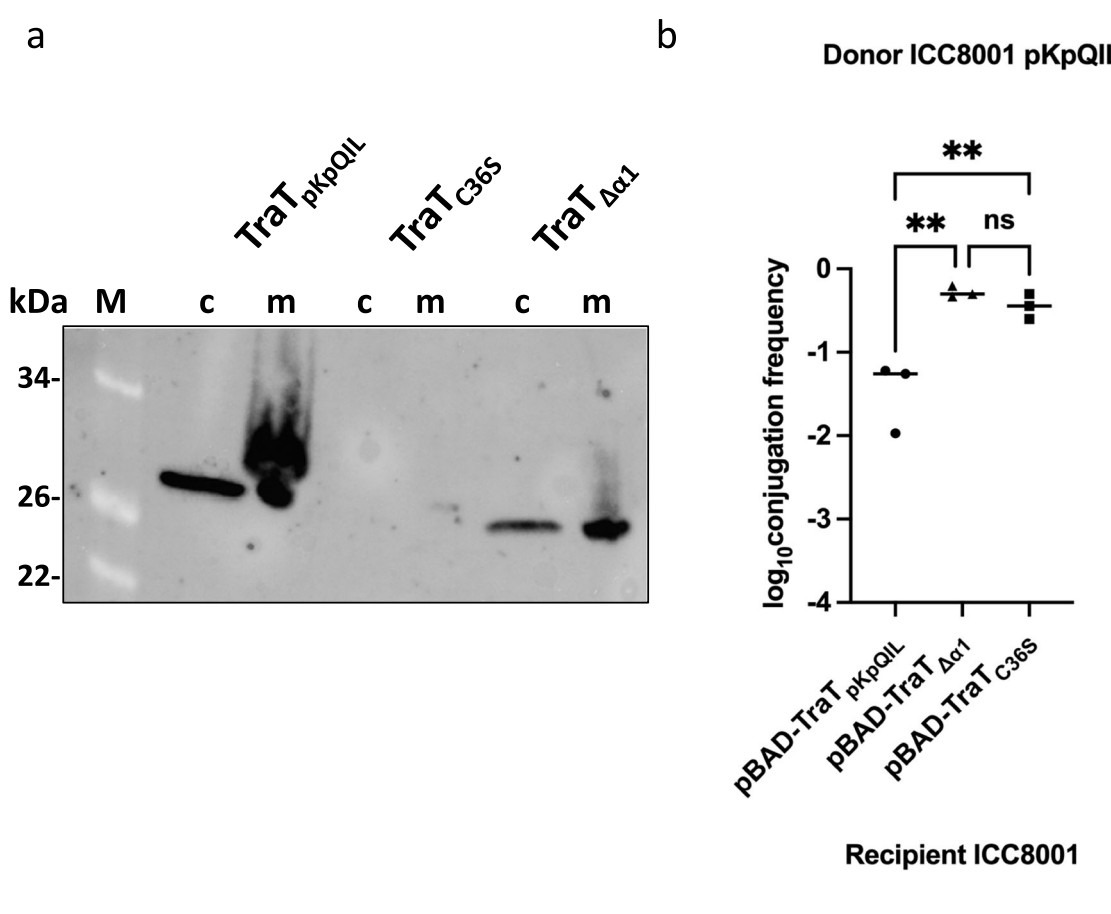

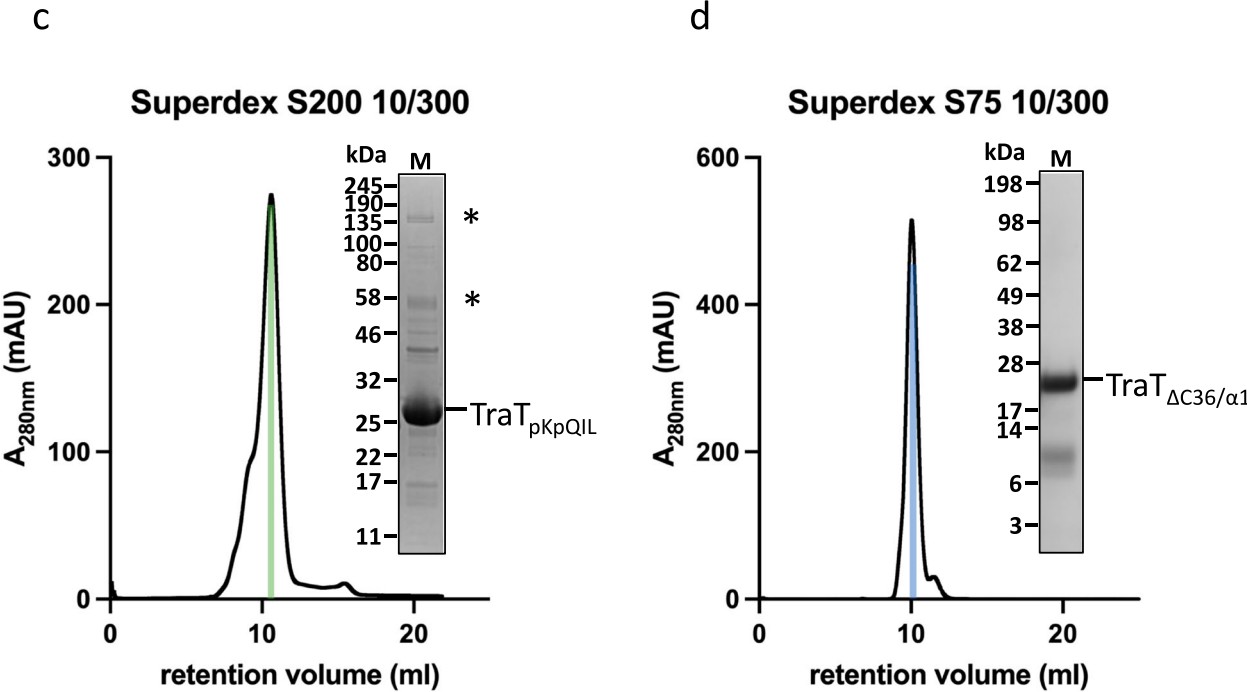

model (LG + G4) identified by ModelTest-NG[28]. In the absence of a known outgroup, the resulting tree was midpoint-rooted (Fig. 6; see https://microreact.org/project/tra-t which includes bootstrap values). Despite the unknown rooting, the phylogeny showed that the chromosomal-encoded sequences, most of which share 30–50% amino acid similarity to $TraT_{pKpQIL}$, were distributed among many highly diverged lineages. Analysis of chromosomal TraT representatives from different lineages, including those identified from *Acidithiobacillus caldus*, *Fusobacterium nucleatum*, *Campylobacter jejuni*, *Nitrosomonas ureae* and *Vibrio ostreae*, showed that the genomic context of the *traT* genes was variable and largely of unknown function (Fig. 7a).

**Fig. 4 | The posttranslationally modified α1 by DAG mediates oligomerisation and insertion to the outer membrane. a** Western blot analysis of the expression of the TraT$_{pQpKIL}$ variants in the OM of *E. coli*; c and m indicate whole cells and membrane fraction, respectively. **b** The TraT variants cannot provide pKpQIL SFX. Significance between TraT$_{pQpKIL}$ and TraT$_{C36S}$/TraT$_{\Delta\alpha1}$ tested by Ordinary one-way ANOVA and Dunnett's multiple comparison test (*n* = 3 biological repeats). TraT$_{pQpKIL}$ vs. TraT$_{\Delta\alpha1}$ *p* = 0.0024, TraT$_{pQpKIL}$ vs. TraT$_{C36S}$ *p* = 0.0051. **c** The mature

full length TraT$_{pKpQIL}$ migrates as an oligomer on a Superdex S200. SDS-stable oligomers are indicated as asterisks. **d** Deletion of both the DAG/PA modified C36 and α1 results in TraT$_{\Delta C36/\alpha1}$ migrating as a monomer on a Superdex S75. No SDS-stable oligomers are observed either. The coloured bars in **c, d** indicate the fraction used for the SDS-PAGE; both panels are representatives from two purifications. Source data are provided as a Source Data file.

Based on AlphaFold 3 structural predictions, the chromosomally-encoded TraTs display high structural similarity to TraT$_{pKpQIL}$ despite the low sequence similarity, as exemplified by TraT$_{Acidithiobacillus}$, TraT$_{Fusobacterium}$, TraT$_{Campylobacter}$, TraT$_{Nitrosomonas}$ and TraT$_{Vibrio}$ (Fig. 7b); the main difference is found within the β-hairpin motif, which has variable length between species that could potentially impact oligomerisation. AlphaFold 3 cannot accurately model the α-helical region, α3 and α4, for all the sequences but it is likely to adopt a similar conformation to the TraT$_{pKpQIL}$ and TraT$_F$ structures (of note, the α-helical region of TraT$_{pKpQIL}$ and TraT$_F$ from AlphaFold 3 is also disordered compared to the cryo-EM structures).

### *traT* has independent origins in Enterobacteriaceae and Legionellaceae F-like plasmids

Our phylogenetic analysis showed that all TraT sequences from IncF plasmids (*n* = 229) found among various Enterobacteriaceae species, including TraT$_{pKpQIL}$ and TraT$_F$, were located within a single plasmid-associated clade (Fig. 6). This clade, comprising a total of 307 sequences, also contained TraT sequences from other Enterobacteriaceae plasmids with no defined replicons. Notably. these included eight plasmids from *Aeromonas spp* and six from Erwiniaceae from which the TraT sequences were found in basal lineages of the plasmid-associated clade. Inspection of these particular plasmid sequences showed that all contained an F-like *tra* operon with the exception of two from *Aeromonas spp* (accessions CP039630 and KX364409).

Analysis of the genetic context of *traT* in different IncF plasmids showed that *traT* is consistently localised downstream of *traS* near the end of the *tra* operon. This was also the case among plasmids with an F-like *tra* operon from *Aeromonas spp* and Erwiniaceae. Together these findings suggest that *traT* genes from these plasmids share a common origin, and that the gene was likely acquired by an ancestral IncF/F-like *tra* operon in a single event that preceded the subsequent diversification of IncF/F-like plasmids in Enterobacteriaceae. We also identified seven TraT homologues in this clade possessing replicon types other than IncF subtypes including Col(pHAD28), IncR and repB. However, all seven plasmids also possessed an F-like *tra* operon with *traT* in the same position as in IncF plasmids, suggesting that they may represent fusion events of different plasmid backbones.

The TraT sequences encoded by the range of IncF plasmids share 71–100% amino acid similarity to TraT$_{pKpQIL}$. We found some clustering of TraT sequences by host genus (Fig. 6). For example, TraT$_F$ and TraT$_{R100}$ (from *E. coli*) belonged to a clade in which the majority (82.6%; 100/121) of sequences were derived from either *Escherichia*, *Shigella* or *Salmonella*, while TraT$_{pKpQIL}$ (from *K. pneumoniae*) was located in a clade in which the majority of sequences (83.3%; 50/60) were from the *Klebsiella*/ *Raoultella* genus. We also found that TraT sequences from plasmids carrying the same replicons usually shared high similarity, such as those with IncFIB(S)/IncFII(S) (97.1–100% identity among 13 plasmids). However, a higher diversity of TraT sequences was found among some replicon types including IncFII(pCoo) (77.0–100% among 12 plasmids) which were found in different sub-clades of the IncF plasmid-associated clade.

We also found that the 14 TraT sequences encoded by plasmids from the Legionellaceae family clustered together into a single clade in the phylogenetic tree, distinct from the Enterobacteriaceae IncF/F-like

plasmid-associated clade (Fig. 6). Thirteen of the sequences were identified among plasmids from *Legionella spp* and one from *Fluoribacter dumoffii*. Of these plasmids, pLPL, found in *L. pneumophila* str. Lens has been shown to be conjugative[29]. Two of the TraT sequences from *Legionella fallonii* were encoded on the same plasmid (accession LN614828) in distinct *tra* operons. The Legionellaceae TraT sequences shared low amino acid similarity (39–44%) with TraT$_{pKpQIL}$. They were also highly diverse among themselves, with pairs sharing 51-100% amino acid similarity, despite grouping together in the tree. Analysis of 11 of the plasmids that were complete or nearly-complete showed that all carried *traT* as part of an F-like *tra* operon, albeit with *traT* at the beginning of the operon upstream of *traA* (Fig. 8a). Overall, the *Legionella* plasmids showed high diversity in terms of gene composition with high divergence also observed within the *tra* operon itself (Fig. 8b). These findings suggest that *traT* was incorporated into the *tra* operon of an early F-like Legionellaceae plasmid, independent of the acquisition of *traT* into Enterobacteriaceae IncF/F-like plasmids and has been maintained within the backbone as these plasmids have subsequently diversified.

In addition to the two plasmid-associated clades described above, three other plasmid-encoded TraT homologues from *Piscirickettsia salmonis*, *Sulfuricurvum kujiense* and *Aromatoleum aromaticum* were found in separate lineages of the tree with each sequence most closely-related to chromosomal TraT sequences from other species. These TraT homologues share 34%, 45% and 38% amino acid similarity, respectively, with TraT$_{pKpQIL}$. In the *P. salmonis* plasmid, p2PS8, *traT* is part of an F-like operon and located near the end of the *tra* operon as in Enterobacteriaceae IncF plasmids. In the *S. kujiense* plasmid, pSULKU01, *traT* was found outside of a *tra* operon although a small number of *tra* gene homologues were identified on the plasmid. In the *A. aromaticum* plasmid, *traT* was also found outside of a *tra* operon although there were a higher number of *tra* genes on the plasmid but with a different organisation to those found in Enterobacteriaceae or Legionellaceae F-like plasmids. Altogether, these findings indicate the occurrence of further independent acquisitions of *traT* by plasmids in these genera.

### Mobilisation of *traT* between plasmids and chromosomes in Enterobacteriaceae

Unlike in the Legionellaceae clade of the phylogenetic tree where TraT homologues were identified only within plasmids, we found that a minority of sequences within the Enterobacteriaceae IncF/F-like plasmid-associated clade were located within chromosomes (Fig. 6). These included a sub-clade of 16 TraT sequences that were exclusively chromosomally-encoded (seven of which are from *Citrobacter spp.*) as well as additional chromosomally-encoded sequences that were sporadically distributed across the clade as singletons or in smaller clusters.

Notably, all the chromosomally-encoded TraT sequences found among this clade were also from the Enterobacteriaceae family. Analysis of the genomic context around *traT* from an *E. coli* ST492 chromosome (strain ED1a; accession CU928162.2) revealed that the gene was encoded on a ~135 kb genomic island (spanning 3,374,882 bp to 3,510,553 bp) which could be identified in only a subset of other ST492 chromosomes, and was found in the absence of a *tra* operon. These findings suggest that *traT* has been occasionally mobilised from IncF/

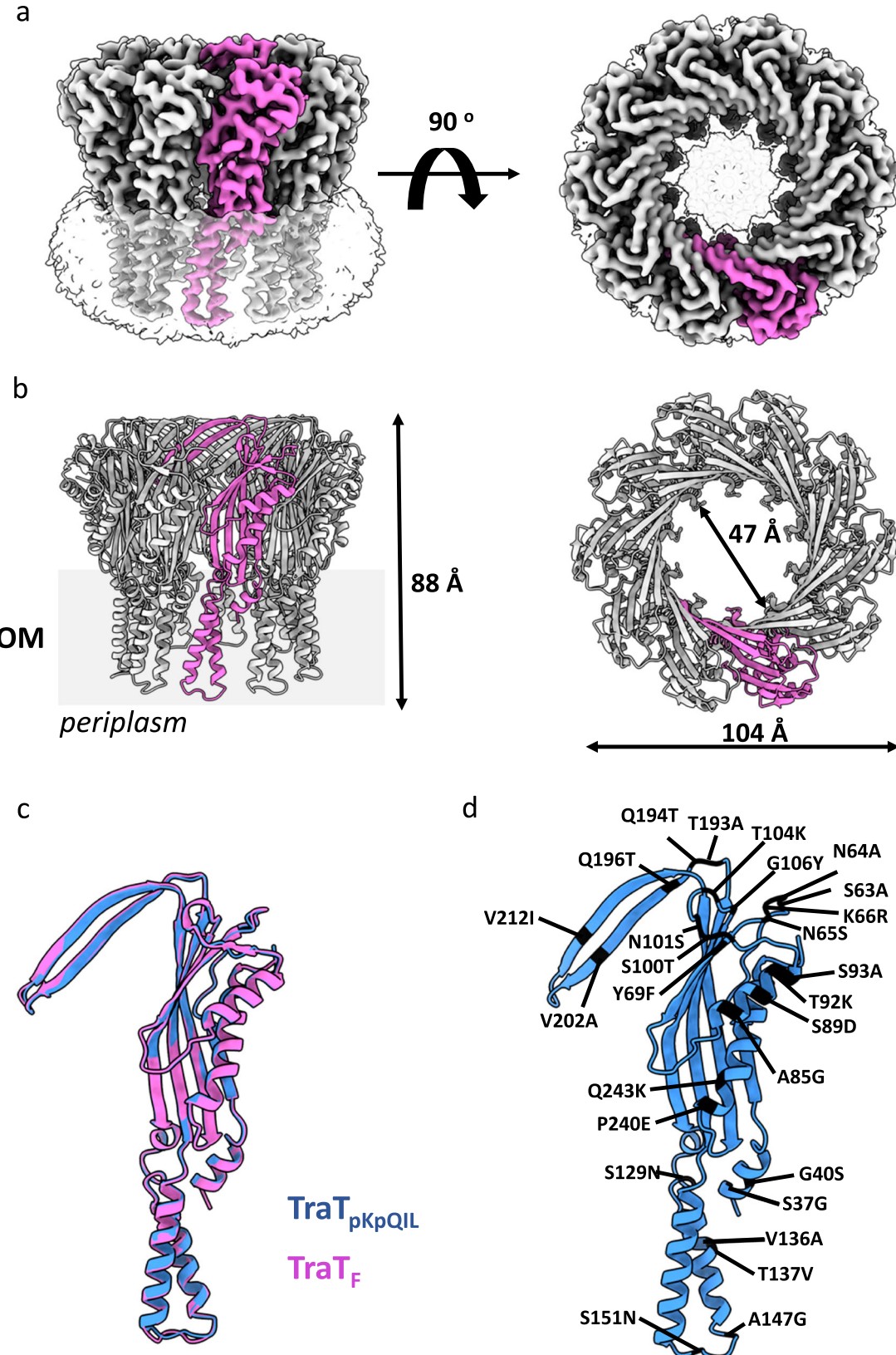

**Fig. 5 | Cryo-EM structure of TraT$_F$. a** Ab initio cryo-EM map of TraT$_F$ at 2.66 Å resolution. A similar ten-fold symmetry to TraT$_{pKpQIL}$ that results in a cork like structure is also observed; the TraT$_F$ protomers are shown in grey and one in pink. The CYMAL-6 micelle is shown as transparent grey. **b** Cartoon representation of the TraT$_F$ oligomer. The structural features of the cork-like structure are similar to the TraT$_{pKpQIL}$ structure. **c** The TraT$_{pKpQIL}$ and TraT$_F$ protomers display very similar conformation. **d** The amino acid differences between TraT$_{pKpQIL}$ and TraT$_F$ have been mapped onto the TraT$_{pKpQIL}$ structure. Most differences are found within the β-sandwich domain suggesting a role in specificity.

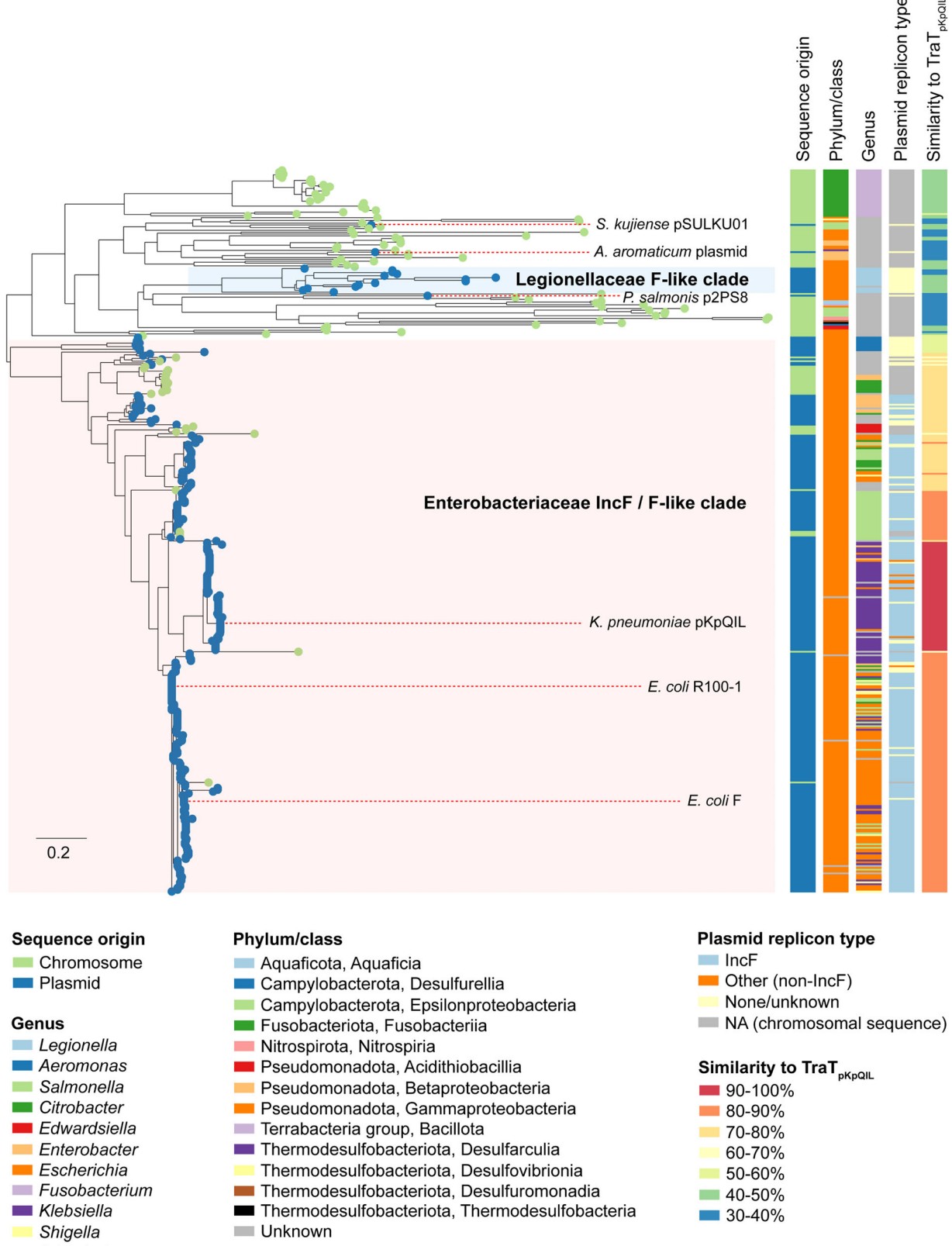

**Fig. 6 | Phylogenetic analysis of TraT.** A midpoint-rooted phylogenetic tree of 399 TraT proteins demonstrates distinct groupings of sequences from *Enterobacteriaceae* IncF/F-like and *Legionella* F-like plasmids among a high diversity of chromosomally-encoded TraT. Isolate tips are coloured according to whether they originate from a chromosomal or plasmid sequence. Metadata columns show the taxonomic group (phylum/class) of the associated host organism and its genus, the plasmid type (if applicable), and the percentage amino acid similarity of each TraT to TraT$_{pKpQIL}$. The scale bar represents the number of substitutions per site. An interactive visualisation of the phylogenetic tree with bootstrap values and associated metadata is available via Microreact: (https://microreact.org/project/tra-t).

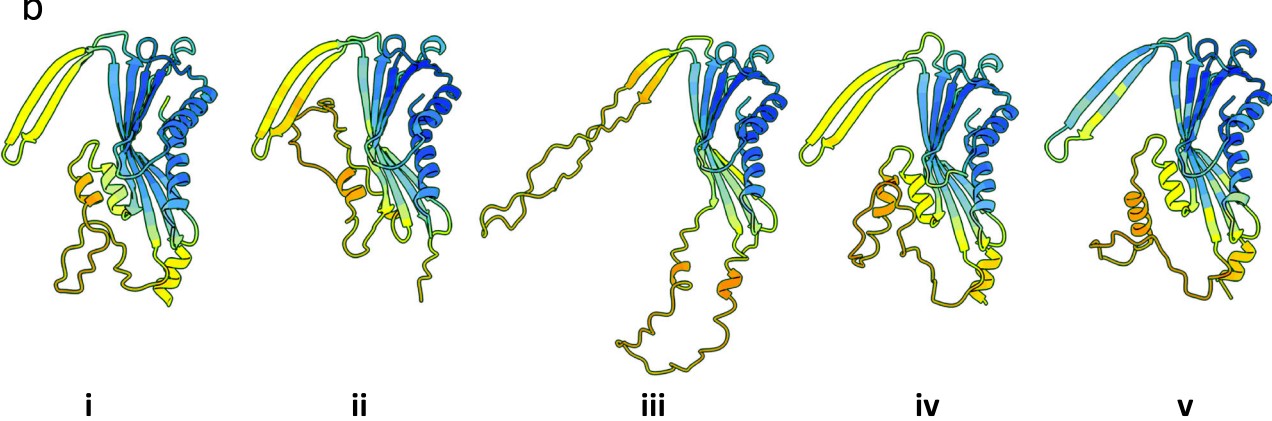

**Fig. 7 | Chromosomal TraTs and their predicted structures. a** Genes flanking the chromosomal *traT* of some representative species (i) *Acidithiobacillus caldus*, (ii) *Fusobacterium nucleatum*, (iii) *Campylobacter jejuni*, (iv) *Nitrosomonas ureae* and (v) *Vibrio ostreae*. TraT is shown in red and the proteins flanking the TraT in grey. **b** AlphaFold 3 predicted structures of the chromosomal TraTs; panels (i–v) correspond to the TraT from the same species as in (**a**). They all display a very similar β-sandwich domain with some variation of the TM helices and β-hairpin motif. Coloured by confidence score (blue, high pLDDT score, to yellow, low pLDDT score).

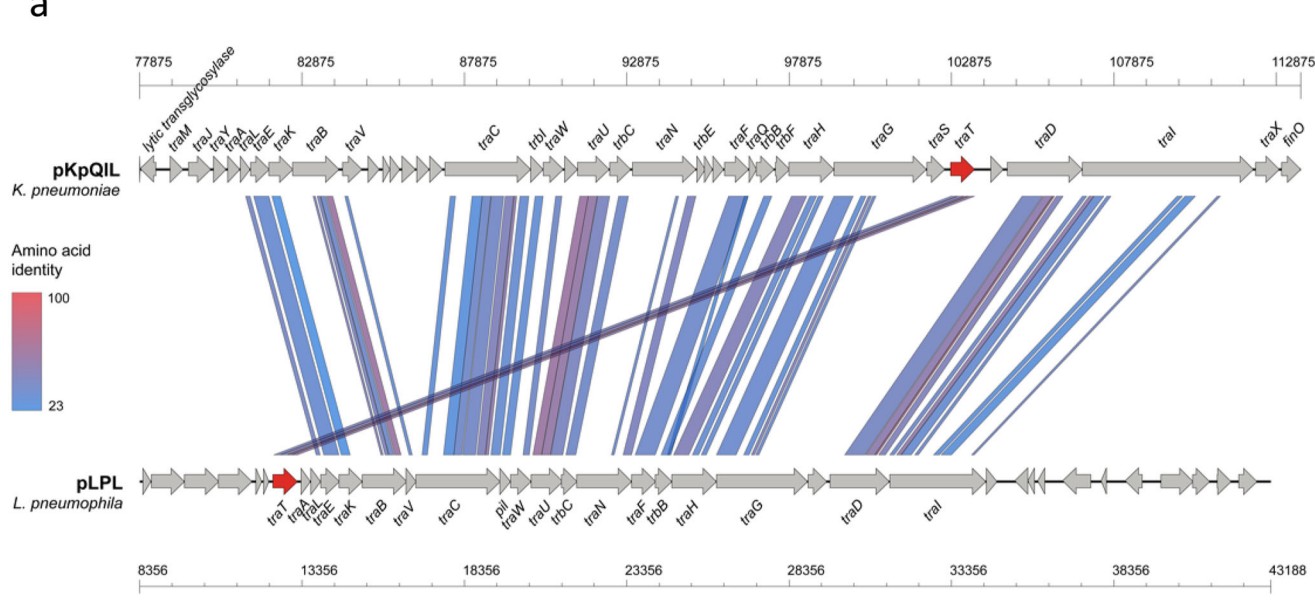

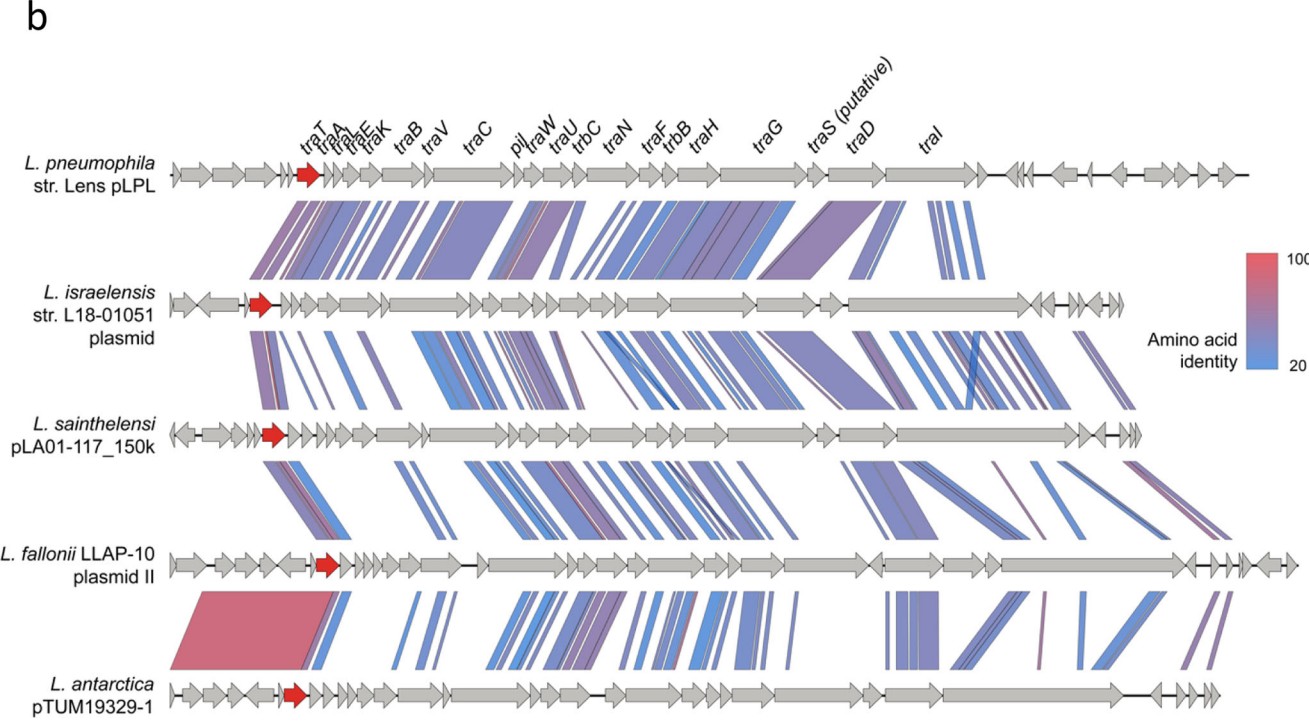

**Fig. 8 | Comparison of F-like *tra* operons. a** Comparison of the *tra* operons from *K. pneumoniae* pKpQIL (accession GU595196) and *L. pneumophila* pLPL (accession CR628339) demonstrates different positioning of the *traT* gene. The coordinates show the respective location within each of the full-length plasmid sequences. **b** Comparison of F-like *tra* operons from diverse *Legionella* plasmids demonstrates conserved positioning of *traT* at the start of the operon.

F-like plasmids onto the chromosome among Enterobacteriaceae species. Analysis of additional chromosomally-encoded *traT* sequences showed that the gene is found within diverse genomic contexts among the different Enterobacteriaceae species.

## Discussion

TraT is a multifaceted OM lipoprotein that has been implicated in various independent cellular functions including immune invasion and SFX. The TraT protein has been shown to inhibit the lysis of sensitised erythrocytes by serum complement, therefore increasing bacterial survival in serum. This inhibition primarily affects the C6 step of the complement cascade, preventing the formation of the membrane attack complex[30]. In *Edwardsiella tarda*, it acts as an anti-complement factor and a cellular infection promoter by binding to complement factor H and inhibiting the alternative pathway of complement activation[31]. This prevents the formation of the MAC, which would otherwise lead to bacterial lysis. TraT can also engage with the host CD46 through a specific domain of the complement control protein, which enhances the infection of cells[31]. TraT has been engineered to present foreign antigens on the cell surface of bacteria as a route to new vaccines[32].

Here, we have probed the role of TraT in the context of SFX and determined its rather unique cryo-EM structure that is capable to provide it with multifunctionality. Once entered a recipient, conjugative plasmids of both Gram-positive and Gram-negative bacteria prevent secondary conjugation events, thus protecting the host from LZ. ENX is widespread, for example via TraS in IncF plasmids, TrbK in RP4 plasmid and Tra130 in the *Enterococcus faecalis* sex pheromone pCF10 plasmid[33]. Here we show that TraT, known for its role in SFX in Enterobacteriaceae IncF plasmids, is also found as a chromosomal gene[18] of unknown function in diverse Gram-negative taxa and has been independently acquired by different plasmid families during its evolution.

In this study we confirmed the plasmid-specific activity of the lipoprotein TraT, as over expression of $TraT_{pKpQIL}$, but not $TraT_F$, specifically inhibited conjugation of pKpQIL. Solving the cryo-EM structures of $TraT_{pKpQIL}$ and $TraT_F$ revealed that both form a similar decameric cork-like structure, which is inserted into the outer membrane via an α-helical barrel domain. The diameter of the decameric structures of $TraT_{pKpQIL}$ and $TraT_F$ is 102 Å, and 104 Å respectively, while the diameter of the inner ring of $TraT_{pKpQIL}$ and $TraT_F$ is 48 Å and 47 Å respectively. We further confirmed that the mature recombinant $TraT_{pKpQIL}$ and $TraT_F$ are posttranslationally modified by DAG and PA[22]. The resolved DAG in the $TraT_{pKpQIL}$ structure is exclusively interacting with residues from TM helices α1, α3 and α4 of the modified protomer. We propose that the role of the DAG is to drive both folding and insertion of TraT to the OM; the AlphaFold 3 prediction of the $TraT_{pKpQIL}$ structure has modelled α3 and α4 as unstructured and with low confidence score, whereas our cryo-EM structures revealed folded helices, suggesting the critical role of DAG to mediate their correct folding and insertion to the OM. Removal of the lipidated α1 resulted in monomeric soluble TraT, mostly due to the destabilisation of the α3 and α4 that form the α-helical barrel. The DAG and PA lipids are essential for the correct processing of TraT to the OM as absence of the modified C36 results in no protein expression.

The 27 amino acids that differentiate $TraT_{pKpQIL}$ from $TraT_F$ have been mapped onto the cryo-EM structures, which are scattered throughout. Interestingly, most of the amino acid differences in the $TraT_F$ are predominantly smaller side chains such as alanine or glycine residues without changing the charge profile of the β-sandwich domain; no changes are found inside the TraT cavity, suggesting that the β-sandwich domain is likely the site of possible interaction with partner proteins from the donor. Although our data together with previous work indicate that MPS and the pilus are not part of the SFX process[21], the subtle amino acid differences between the TraT sequences provide the high degree of specificity. It is likely that specificity is driven by either the α2 or/and β-hairpin motif as the amino acids display low conservation (Supplementary Fig. 5). Moreover, TraT might be interfering with post-MPS steps, and associated proteins, that are currently unknown. Accordingly, the molecular basis of SFX remains indefinable but our data points to the role of the extracellular domain to mediate this process.

Identification and analysis of homologues from the UniprotKB database unexpectedly revealed that TraT sequences are also encoded on chromosomes with a distribution across multiple bacterial phyla. The chromosomal encoded TraT sequences form many divergent lineages in the phylogenetic tree. In addition, the chromosomal TraT sequences are flanked by genes encoding different functions such as unrelated enzymes or hypothetical proteins, whose role is not related to SFX. Taken together, these findings suggest that *traT* may originally have evolved as a chromosomal gene, despite its more familiar role in SFX in the *Enterobacteriaceae* IncF plasmid *tra* operon. We found that the overall AlphaFold 3 predicted structures of chromosomal TraT sequences are similar to the plasmid-encoded TraT sequences; the main difference is found within the β-hairpin motif, which has variable length between species that could

potentially impact oligomerisation. While TraT has been shown to exhibit serum resistance activity, its identification in environmental bacteria (e.g. Nitrosomonas) suggests that this might not be the main selective pressure[23]. While chromosomal TraT sequences might provide these bacteria with a SFX activity, protecting them from fitness costs from incoming plasmids, further studies are needed to determine their precise role.

The plasmid-encoded TraTs cluster in distinct clades of the phylogenetic tree, largely belonging to two separate lineages comprising TraT from Enterobacteriaceae IncF/F-like plasmids and Legionellaceae F-like plasmids. These findings suggest that TraT was acquired by the plasmid backbone of an early ancestor from each of these plasmid families on independent occasions. The different positioning of *traT* within the *tra* operons of these different plasmid families further supports independent acquisitions. We also detected *traT* genes in individual plasmids from *P. salmonis*, *S. kujiense* and *A. aromaticum*, which were found separately among distinct lineages of the phylogenetic tree. This is suggestive of further independent acquisitions of *traT* by plasmids circulating in these taxa. Notably, however, *traT* from the *P. salmonis* plasmid, p2PS8, is located within the same position of an F-like *tra* operon as in the Enterobacteriaceae IncF plasmids, despite its seemingly independent origin. Further work will be required to understand whether TraT plays the same role among the different plasmid families. In the Enterobacteriaceae, our identification of *traT* sequences across the IncF/F-like plasmid-associated clade that are encoded on chromosomes further demonstrates the dynamic mobilisation of this gene, and may also reflect the functioning of TraT in multiple unknown roles within the bacterial cell.

In conclusion, our study highlights the role of TraT, a unique outer membrane lipoprotein, in SFX during plasmid conjugation. Through cryo-EM structural analysis, we elucidated the distinct architectures of $TraT_{pKpQIL}$ and $TraT_F$, revealing their posttranslational modifications and their importance in proper membrane insertion and oligomerisation. The specificity of TraT's function in SFX appears to be influenced by subtle amino acid variations and potentially involves interactions with host proteins, though the precise mechanisms remain to be fully defined. Moreover, our bioinformatics analysis uncovers the widespread distribution and evolutionary history of *traT*, indicating that it may have originally emerged as a chromosomal gene before being incorporated into diverse plasmid families through independent acquisition events. This genomic plasticity not only suggests potential roles for TraT beyond its well-characterised functions but also emphasises its adaptive significance in various bacterial taxa.

## Methods

### Generation of TraT constructs

The full length sequence of $TraT_{pKpQIL}$ (accession ID: ARQ19738.1) and $TraT_F$ (C21-L244) (accession ID: WP_000850422.1) were PCR amplified from KP pKpQIL-UK and the pOX38 plasmids, respectively. The PCR fragments were subcloned into the pET28b expression plasmid or a modified pBAD plasmid that carries KAN resistance. The cloned *traT* gene is followed by a TEV protease cleavage site and a C-terminal His6-tag. The pET28b-$TraT_{pKpQIL}$ construct carries a A210T mutation from cloning. The $TraT_{\Delta C36/\alpha 1}$ (cloned sequence: E49-L258) was PCR amplified from the pKpQIL-UK plasmid and subcloned into the pET28b plasmid.

The $TraT_{C36S}$ and $TraT_{\Delta\alpha 1}$ (cloned sequence: C36 + E49-L258) mutants (in pBAD and pET28b backgrounds) were generated by site directed mutagenesis. Polymerase chain reaction (Q5 2x MasterMix, NEB) was followed by a one-step kinase, ligase and Dpn1 reaction (KLD Enzyme Mix, NEB) and transformation into chemically competent DH5α cells. Constructs were verified by sequencing (Eurofins Genomics).

### Selection-based conjugation assay

The bacterial strains used for selection-based conjugation assays are listed in Supplementary Table 2. In conjugation experiments, where *K. pneumoniae* was used as the recipient, the TraT constructs in the pBAD vector were transformed into ICC8001 using a room-temperature electrocompetent cell protocol as previously described[34]. The conjugation assays were performed as previously described using the previously generated sfGFP-expressing pKpQIL reporter plasmid pKpGFP[17]. In brief, 1 mL aliquots of overnight cultures of donor bacteria carrying pKpGFP and recipient bacteria were pelleted by centrifugation at $5000 \times g$ for 5 min. Following resuspension in 1 mL PBS, donor and recipient bacteria were mixed at an 8:1 v/v ratio. The conjugation mixture was plated onto an LB agar plate containing 0.5% L-arabinose and incubated at 37 °C for 6 h. The resultant conjugation spot was resuspended, and serial dilutions were spotted in triplicate onto a selection plate to select for and quantify the number of recipients. Recipient colonies were selected on kanamycin-containing LB agar plates and transconjugants were selected on LB agar plates containing kanamycin and ertapenem. Plasmid uptake in transconjugant colonies was confirmed through visualising GFP fluorescence on a Safe Imager 2.0 Blue Light Transilluminator (Thermo Fisher). The colony forming units per mL (CFU/mL) were determined for both the number of recipients and the number of transconjugants. The conjugation frequency was calculated by dividing the CFU/mL of transconjugants by the CFU/mL of recipients. The data was log base 10 (log10) transformed, followed by statistical analysis in GraphPad Prism.

### Overexpression of recombinant TraT proteins

Constructs were transformed into *E. coli* C41 (DE3) competent cells (F− ompT gal dcm hsdSB(rB- mB-)(DE3))[35] and expressed in LB medium supplemented with 50 μg/mL kanamycin. A single colony was used to inoculate 200 mL LB media, supplemented with the relevant antibiotic(s), and incubated at 37 °C with orbital shaking at 200 rpm for 16–18 h. Ten mL of preliminary culture was used to inoculate 1 L LB media supplemented with 50 mg/ml kanamycin. Cultures were grown at 37 °C to an $OD_{600}$ of 0.6 then induced with 1 mM of isopropyl β-d-1-thiogalactopyranoside (IPTG) and maintained at 37 °C for 3 h.

### TraT$_{pKpQIL}$ and TraT$_F$ purification

TraT was purified from OM vesicles (OMVs) that were prepared as previously described[36]. OMVs containing TraT$_{pKpQIL/F}$ were solubilised with 1% (w/v) CYMAL-6 (Anatrace) at 4 °C with 150 rpm stirring for 1 h. Insoluble material was removed by ultracentrifugation at $131,000 \times g$ for 1 h at 4 °C. The supernatant was combined with 30 mM imidazole then loaded onto an Econo-Column (Bio-rad) containing 5 mL Ni-NTA resin at 4 °C. IMAC was performed, with 10 column volume (CV) washes of IMAC buffer containing 30 mM imidazole followed by the elution of TraT$_{pKpQIL/F}$-His$_6$ in IMAC buffer containing 250 mM imidazole. The elute was proteolytically cleaved with His$_6$-tagged TEV protease overnight at 4 °C at a TraT-to-TEV ratio of 1:1, whilst dialysing against dialysis buffer. Proteolytically cleaved TraT$_{pKpQIL/F}$ was passed over an Econo-Column containing 5 mL Ni-NTA resin and was collected in the FT. TraT$_{pKpQIL/F}$ was further purified by SEC with a Superdex S200 10/300 column (Cytiva) using an ÄKTA pure system (Cytiva). Sample purity was assessed by SDS-PAGE.

### Western blot analysis

Overexpression of the TraT$_{pKpQIL}$ variants was performed as above. Their expression in the OM of *E. coli* was tested by Western blot analysis. The antibodies were 6x-His tag monoclonal (HIS.H8) (Invitrogen) and Rabbit anti-mouse IgG, HRP (Invitrogen). The membranes were stained with the chemiluminescent Pierce™ ECL Western Blotting Substrate (Thermo Fisher Scientific) and the detection of horseradish peroxidase signal on the immunoblot was visualised using a Chemi-Doc™ MP Imaging System (Bio-Rad).

### TraT$_{\Delta C36/\alpha1}$ purification

Cell pellets were resuspended in 1 X PBS containing 90 U/ml Benzonase Nuclease (Sigma), 5 mM MgCl$_2$ and 0.5 mg/ml Pefabloc®. The cell resuspension was passed through a cell disruptor twice at a process pressure of 28 kpsi. Soluble matter was separated from unbroken cells and membranes via ultracentrifugation at $131,000 \times g$ for 1 h at 4 °C. The supernatant was combined with 50 mM imidazole and passed over a 5 mL His-Trap column. IMAC was performed, where the column was washed with 10 CVs of IMAC buffer (1 X PBS supplemented with 50 mM imidazole) followed by the elution of TraT$_{\Delta C36/\alpha1}$-His$_6$ in IMAC buffer containing 250 mM imidazole. The eluate was proteolytically cleaved with His$_6$-tagged TEV protease overnight at 4 °C at a TEV-to- TraT$_{\Delta C36/\alpha1}$ ratio of 1:10. Proteolytically cleaved TraT$_{\Delta\alpha1}$ was passed over a 5 mL His-Trap column and it was collected in the FT. The TraT$_{\Delta C36/\alpha1}$ oligomeric state was assessed by SEC using a Superdex S75 10/300 column using an ÄKTA pure system. Sample purity was assessed by SDS-PAGE.

### Cryo-EM grid preparation and screening

Cu300 mesh 1.2/1.3 holey carbon grids were glow discharged using a GloQube Plus Glow Discharge System for 30 s at 30 MA. Grids were loaded onto a Vitrobot Mark IV (FEI Thermo Fisher) operating at 4 °C with 100% humidity. Four μL of sample at 4–5 mg/mL was applied and blotted to the grid with a blot force of 3 and a blot time of 3 s. The grids were flash frozen in liquid ethane.

### Cryo-EM data collection

TraT$_{pKpQIL}$ and TraT$_F$ datasets were collected at the electron Bio-Imaging centre (eBIC) on a 300 kV FEI Titan Krios EM (Thermo Fisher Scientific). The specifications of the microscope and the parameters of the respective datasets collected are listed in Supplementary Table 1.

### Cryo-EM data processing

Cryo-EM datasets were processed using CryoSPARC v4.0.3[37]. A total of 6290 movies from the TraT$_{pKpQIL}$ dataset and a total of 2460 movies from the TraT$_F$ dataset were imported and corrected using patch motion correction. The corrected micrographs underwent patch CTF, then automatic particle picking was performed using 'Blob picker', with minimum and maximum particle diameters of 100 Å and 200 Å respectively. Blob picker identified 2,449,781 TraT$_{pKpQIL}$ particles and 1,109,133 TraT$_F$ particles. Using the 'Inspect Particle Picks' tool, 908,660 TraT$_{pKpQIL}$ particles and 273,302 TraT$_F$ particles were extracted using a box size of 440 px. The extracted particles underwent 2D classification resulting in 9 classes of TraT$_{pKpQIL}$ (448,485 particles) and 13 classes of TraT$_F$ (131,705 particles). Using the selected classified particles, a 3D reconstruction was generated using the 'ab-initio reconstruction' job, which used 210,600 of the TraT$_{pKpQIL}$ particles and all 131,705 of the TraT$_F$ particles. For both TraT$_{pKpQIL}$ and TraT$_F$, C1 symmetry was initially applied, which showed the presence of a 10-fold symmetry. Local CTF refinement was performed and a high-resolution density map was generated by homogenous refinement, where C10 symmetry was applied to both TraT$_{pKpQIL}$ and TraT$_F$ densities, resulting in resolutions of 2.72 Å and 2.92 Å respectively. To further improve the resolution of the maps, the output half maps were used to perform local CTF refinement. The resultant particles with updated CTF parameters underwent a second iteration of homogenous refinement with C10 symmetry, resulting in finalised maps of TraT$_{pKpQIL}$ at 2.47 Å and TraT$_F$ at 2.66 Å.

### Model building, refinement and validation

An initial model of monomeric TraT$_{pKpQIL}$ was generated by Buccaneer within the collaborative computational project for electron cryo-microscopy (CCP-EM) suite[38]. The model went through multiple iterations of refinement in Phenix using *phenix.real_space_refine*[39]. Density for a partially resolved DAG molecule was visible in the

TraT$_{pKpQIL}$ map and was manually placed in Coot[40]. The *apply_ncs* function was used in Phenix to generate the TraT$_{pKpQIL}$ decamer which was further refined and validated with MolProbity[39,41]. For the TraT$_F$ structure, the finalised cryo-EM monomeric model of TraT$_{pKpQIL}$ was docked into the TraT$_F$ map in Phenix using *phenix.dock_in_map*. The model went through multiple iterations of refinement in Phenix using *phenix.real_space_refine*[39]. A decamer of TraT$_F$ was generated and corrected by further rounds of refinement. *apply_ncs* was used to generate the TraT$_F$ decamer which was further refined and validated with MolProbity. Only weak density was observed for DAG in the TraT$_F$ map and it was not included in the final model.

**Phylogenetic analysis of TraT**
TraT homologues were identified using UniprotKB by searching for "gene=*traT*" and "protein=TraT". Sequences were filtered to include those of 200–300 amino acids in length and possessing ≥30% amino acid identity to TraT$_{pKpQIL}$. Furthermore, only sequences with a defined chromosomal or plasmid origin were included, as determined from the associated sequence accessions, largely resulting in the exclusion of those identified from short-read genome sequencing data.

The filtered protein sequences of TraT were aligned using Clustal-Omega v1.2.4[42]. ModelTest-NG[28] was then used to compare evolutionary models applied to the protein alignment. A maximum-likelihood phylogenetic tree was constructed using the best-fitting model (LG + G4) using RAxML-NG v1.2.0[43] with 1000 bootstrap replicates. Microreact was used for visualisation of the phylogenetic tree with associated metadata[44].

**Genomic context analysis of TraT**
All plasmid sequences carrying plasmid-encoded TraT sequences were downloaded from public sequence archives using the available sequence accessions identified via UniprotKB. Plasmid replicons were identified from these sequences using PlasmidFinder v2.0[45].

The genomic context of *traT* genes in different plasmids was compared with Genofig v1.1.1[46] using tblastx to perform homology searches. Regions of homology were determined using a minimum length of 20 nucleotides and a minimum identity of 20%.

**Reporting summary**
Further information on research design is available in the Nature Portfolio Reporting Summary linked to this article.

## Data availability
The cryo-EM maps have been deposited in the Electron Microscopy Data Bank (EMDB) under accession codes EMD-50728 (TraT$_{pKpQIL}$) and EMD-50723 (TraT$_F$). The structural coordinates have been deposited in the RCSB Protein Data Bank (PDB) under the accession codes 9FSM (TraT$_{pKpQIL}$) and 9FS5 (TraT$_F$). Raw data for TraT$_{pKpQIL}$ and TraT$_F$ were submitted to Electron Microscopy Public Image Archive (https://www.ebi.ac.uk/pdbe/emdb/empiar/) with IDs EMPIAR-12468 and EMPIAR-12465, respectively. The interactive phylogenetic tree can be accessed from: https://microreact.org/project/tra-t. Source data are provided with this paper.

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

## Acknowledgements

We would like to acknowledge Diamond for access and support of the cryo-EM facilities at the UK national electron Bio-Imaging Centre (eBIC), proposal BI25127. We would like to thank the BSRC mass spectrometry and proteomics facility at the University of St Andrews for mass spectrometry analysis. C.S. was funded by a BBSRC DTP Studentship grant (BB/M011178/1; K.B. and G.F.). N.I. is supported by a fellowship from The Naito Foundation, Japan. G.F. is funded by a grant from the Wellcome Trust ((107057/z/15/z and 224282/Z/21/Z). S.D. is funded by the Bill & Melinda Gates Foundation (investment number INV-025280).

## Author contributions

K.B. and G.F. designed and managed the overall project. C.S. performed TraT purification, conjugation assays, cryo-EM data collection and analysis. N.I. performed TraT mutant purification and analysis. W.W.L., J.L.C.W., S.H., and J.B. prepared the plasmids for conjugation assays. C.S. and K.B. built and refined the structures. G.F. supervised the conjugation assays. S.D. performed the bioinformatics analyses. All authors contributed to the manuscript preparation.

## Competing interests

The authors declare no competing interests.
