## [Transparent Peer Review file · Nature Communications]

Cryo-EM structure and evolutionary history of the conjugation surface exclusion protein TraT

Corresponding Author: Dr Konstantinos Beis

Version 0:

Reviewer comments:

Reviewer #1

(Remarks to the Author)

The authors report that the surface exclusion protein TraT, encoded by the F plasmid, assembles as a decameric ring that protrude across the outer membrane of Gram-negative bacteria. Unlike other OMPs, however, the outer membrane-spanning portion of the TraT ring is composed of alpha-helices, not b-barrel strands. Additionally, the bulk of the ring, which does adopt a b-barrel channel configuration, protrudes from the cell surface. Interestingly, the authors identify TraT homologs among many other F-like plasmids found in Enterobacteriaceae and Legionella spp., as well as non-F-like plasmids and chromosomes across multiple Gram-negative phyla. Thus, while F-encoded TraT plays a minor role in blocking redundant transfer of F plasmids in donor cell populations, the findings suggest these proteins have evolved other presently unknown functions of importance in various infection and environmental settings. Overall, the findings add new insights into the factors that contribute to F plasmid dissemination and further supply evidence that an unusual bacterial surface structure has been exapted over evolutionary time for other biological applications. The comments below are intended to strengthen an already interesting manuscript and allow for more facile comparisons with other TraT homologs.

Major:

1. Numbering of TraTpKpQIL protein sequence. I spent some time trying to figure out the numbering scheme the authors used for TraTpKpQIL based on the published sequence of plasmid pKpQIL (NC_014016.1). For some reason, the authors accessed a sequence (ARQ19738.1) that most likely incorrectly starts with a Met codon that is in frame with the start codon used for all other TraT homologs but resides 15 codons upstream. This yields an unusually long signal sequence of 35 residues, but more problematically a pre-protein of 258 residues as opposed to 243 residues in length - the lengths of other TraT homologs such as F that is also analyzed in this study. Upon processing at the conserved Cys residue, these TraT homologs are both 243 residues. Yet, by adopting the upstream Met as the (incorrect) start codon, TraTpKpQIL's residue numbers don't match those of TraT and most other TraT homologs, which complicates direct comparisons. I suggest two solutions. First, start TraTpKpQIL numbering at the downstream Met codon, so the pre-protein has 243 residues like other homologs. Second, include an alignment of TraTpKpQIL and TraTF, as well as an alignment of these proteins with a selection of other plasmid- and chromosomally-encoded TraT's. The value of these alignments is that they identify regions of variation among the homologs; having done this myself, it is evident that they cluster in only a few locations that can then be readily identified in the solved or AlphaFold-predicted TraT structures – these locations might mark functionally important sites that confer system specificity.

2. L. 144. This truncation is testing for the combined importance of the DAG modification of C36 and alpha helix 1, not just the DAG modification, on TraT oligomerization in vitro. Although the tested mutation abolishes oligomerization, it cannot be concluded whether this is due to the loss of DAG modification, alpha helix 1, or both. To this end, a i) simple substitution of C to S and ii) a deletion of alpha helix 1 but with retention of C36 also should be made and tested for effects on TraT oligomerization in vitro. Additionally, with these mutations in hand, the authors should evaluate their in vivo effects on TraT SFX activity in matings by expression of the mutant proteins in recipient cells. This would allow for firm conclusions concerning the functional importance of lipid modification (OM anchoring) and the alpha helix 1 on protein activity in vivo.

3. Discussion. This could be strengthened considerably by:

i) Rewriting the first paragraph, which is pretty weak. Suggest highlighting the broader functional importance of TraT in SFX, serum resistance, resistance to lytic action of complement, as well as its early use for surface display of antigens of interest. There were also early reports that traT genes are widespread on F plasmids as well as other virulence plasmids

independently of association with tra genes - see Micro Rev. 54:331.

ii) Reducing the extent to which the Results are simply restated, and expanding on the broader biological significance of the findings.

iii) Last paragraph - Although the parallel roles of TraT and EspZ in blocking redundant transfer events are interesting, the mechanisms may be quite distinct – and the biological consequences clearly are (one blocking redundant transfer of a plasmid among bacteria, the other preventing transfer of too many effectors to a mammalian cell). As a last paragraph, however, the comparison distracts from the novelty and main findings of the present work, especially since it ends with the unfortunate conclusion that we still have no understanding of underlying mechanism in either instance. Suggest ending the discussion with a stronger paragraph that highlights the novelty/significance of the present work in the context of TraT's role in plasmid biology and ways this surface protein may contribute more broadly to infection or survival in the environment.

Minor:

4. Fig. 1b. Although it is not mentioned in the legend, I assume that the line represents the overall average of the data set being shown. If so, the line representing the transfer frequency into a recipient carrying pBAD-TraTF is aberrantly high, being in the middle of the two higher data points apparently without factoring in the lower data point.

5. L. 156. Fig. 5D. This is why the numbering should be made uniform – the numbering for residues in fig. 5D refers to TraTpKpQIL, forcing the reader to subtract 15 residues to identify the corresponding residues in TraTF.

6. L. 169. Fig. 5. Panel C plus a sequence alignment would more clearly depict the variation between the two TraT's; by just showing Panel C, one cannot see which variations are conservative and which affect charge, hydrophobicity or size that might confer system specificity. The superimposition overrepresents the differences, which for the most part are highly conservative, e.g., G-S, S-A, T-A, V-A, K-R. The few differences of possible functional importance, e.g., at residues 93 (T-K), 105/107 (T-K, G-Y), 240/243 (P-E, Q-K), could be highlighted and, ideally, tested through mutational analyses (although this is not essential for the present manuscript).

7. L. 196. The mature forms of TraTF and TraTR100 are 100 % identical – the single amino acid difference is in the signal sequence.

8. L. 240. Direct the reader to Fig. 6C for the AlphaFold3 structures referred to in this sentence.

9. L. 267. This would be more clearly presented with an alignment that highlights the few noteworthy differences in charge, hydrophobicity or charge (also, I count only 25 differences but I might be wrong).

10. L. 287. Add the citation here.

11. Supplementary Fig. 4 legend – appears to be copy-pasted from Suppl. Fig. 2 - Change references of "TraTpKpQIL" to "TraTF."

12. Supplementary Table 2. GFP-D – it's mentioned that pKpQIL and sfGFP controlled by Plac. What does this mean? Assuming that sfGFP expression is controlled by Plac, does Plac control something else on pKpQIL, e.g., transfer? Explain. Also, for rigor/reproducibility, the source should be a prior publication, not an author's name. This is also valid for pBAD-traTpKpQIL – was this constructed here or was it published previously, those are the two options, not an author's name as the source. Finally, the more conventional designation is Ery, not Ert, assuming this is the abbreviation for erythromycin.

Reviewer #2

(Remarks to the Author)

Focusing on surface exclusion, the authors investigate the structure, taxonomy distribution, and phylogeny of protein TraT. In their experiments, two TraT variants, TraT_{pKpQIL} and TraT_F, show plasmid surface exclusion specificity. They resolved the structure and confirmed that DAG modified the structure, which cannot be predicted using AlphaFold 3. By searching in the UniProtKB database, the authors found TraT homologs in a variety of taxa, on both chromosomes and plasmids. Phylogeny and comparative genomic content analyses reveal that plasmid TraT variants have different evolutionary histories, largely belonging to two groups.

The authors began with a very specific aspect of the plasmid conjugation process, presenting clear evidence in Figure 1. However, neither the subsequent structure experiments nor the genomic analysis clearly explained the initial observation. Instead, the genomic analysis suggests independent acquisitions of TraT in two taxonomic groups, making this manuscript appear imbalanced and fragmented.

Since I am not an expert in solving protein structures, below are my concerns and questions mainly regarding the genomic analysis.

Line 26-28, 86-87: The phylogenetic analysis is disconnected from the former results, lacking a clear motivation.

Line 28-29: this sentence interrupts the structure. The author should first hypothesize what was expected. For example, TarT on chromosomes has never been reported.

Line 182: Before phylogeny reconstruction, please control the alignment quality. Sequences producing low alignment quality can be observed in Figure 6a, corresponding to unusually long branches. Please add bootstrap support and indicate the protein evolution model and rooting method. The author should consider cutting the tree by above 65% identity to TraTpKpQIL, long branch effects have a detrimental impact on phylogeny reconstruction and rooting. When the conclusion is the independent evolution of TarT in Enterobacteriaceae and Legionella, Figure 6a should mainly display this result.

Line 195: "Some clustering" is imprecise. I suggest reporting the percentage of TraT variants found in each sub-lineage (clade) on the phylogenetic tree. The author should perform constrain tree topology tests on clustering/ lineage.

Line 199-202: The reported replicon info (Col(pHAD28)) here is not precise, PlasmidFinder identifies marker sequence without considering the plasmid backbone. pSTN0717-36-2 (accession: AP022496) is a 118,786 bp plasmid and encodes a F-like repA1 gene (locus tag: STN0717CIT36_P20010). This plasmid likely results from a plasmid fusion event (a small size ColE1-like plasmid and a large size F-like plasmid), similar to the multiple-replicon plasmids reported in the interactive tree display (Col156, IncFIA, IncFIB, IncFII; ColRNAI). The repB replicon is not a standard replicon type.

Line 203; 220; 228: Expressions like a clade/ a sub-clade/ this clade, are not very reader-friendly, readers have to figure these clades out on the interactive tree display, with only figure 6a, it's impossible.

Line 209-212, 228-235: This part of comparative analysis is of special interest. It seems like tarT is mobile via unknown means (or assembly artifacts). This should be discussed based on observations from the flanking sequences.

Line 211-212: I manually checked the functional annotation of the flanking protein-coding genes in plasmid pPC1000_1 (accession CP059403), there is a tar-like operon from tarD (locus tag: FOLKNPGA_03766) to tarT (locus tag: FOLKNPGA_03787). The protein sequences may be quite different from the other five tar operons in Figure S5.

Line 223-227: I found the expression "additional" here confusing. To me, "additional" refers to traT duplications (in chromosomes and plasmids).

Line 231-232: Please provide the sequence location/reference for this genomic island.

Line 236: The high structural similarity is AF3-predicted. This should be clarified.

Line 283-284: If a phylogenetic root is identified in the chromosomal clade on the tree, the suggestion here could be more trustworthy. (Note: this should be tested)

Line 65: refractory -> refractory

Reviewer #3

(Remarks to the Author)

In the manuscript under consideration, Seddon and colleagues investigate the activity and structures of two proteins that inhibit plasmid uptake via surface exclusion. Data are presented for two homologs, both known as TraT, from *Klebsiella pneumoniae* (TraTpKpQIL) and the F plasmid (TraTF). Intriguingly, the two proteins do not appear to be compatible as TraTF cannot recapitulate the surface exclusion activity of TraTpKpQIL in vivo. However, the two proteins form remarkably similar structures as determined by cryogenic electron microscopy leading to the hypothesis that small changes in sequence account for this differential activity. Although the authors present technically sound structures, the manuscript does not fully utilize these structures to interrogate the underlying mechanism of surface exclusion. Included below are a series of suggestions that would strengthen the manuscript and broaden its impact.

Major Considerations

1. The negatively charged ring along the outer rim of the extracellular face of TraTpKpQIL is interesting though it is not clear if this feature is conserved in TraTF. The authors should generate an electrostatic map of TraF for comparison. Similarly, residues within this feature should be mutated and the SFX efficiency evaluated using the conjugation assays described. The authors note that a TraTpKpQIL mutant missing helix 1 is no longer able to oligomerize in vitro and attribute this function to DAG binding. The authors should mutate residues within the DAG binding site and the modified cysteine residue and perform similar SEC experiments to illustrate the role of DAG in oligomerization.

A list of substitutions are plotted on the TraTpKpQIL structure in Figure 5D which appear to be scattered about the structure. However, because TraT is forming oligomeric structures, it is possible that these substitutions map to three-dimensional sites on the decamer and not the monomer. The authors should generate a figure showing these substitutions on the deacmer structure.

The authors have assembled 288 TraT sequences and found some divergent sequences. The conservations of residues found in these sequences should be mapped to the structures of TraT reported here to illustrate variability across species and genetic locus (plasmid vs. chromosome).

The formation of the decamer structure is intriguing but no functional rationale is presented as to why TraT must form these oligomeric structures to function. Similarly, it is not clear if the monomer is incapable of performing SFX. The authors should evaluate the efficiency of TraTpKpQIL monomeric mutants in conjugation assays.

Minor Suggestions

Line 49: Grammatical error – “we have recently showed” should be “we have recently shown”

Line 103: The oligomeric structure should be referred to as decameric.

Line 106: The authors refer to the profile of TraTpKpQIL over a SEC column but do not refer to the data which is presumably included in Figure 4. The authors should reference this result.

Line 116: Grammatical error: “displays a 10-fold symmetry” should be changed to “displays 10-fold symmetry”.

Line 119: The authors indicate that a belt of featureless density surrounds the helical segment of TraTpKpQIL but do not indicate if additional density is found within the helical belt. Without clarification it is not clear if TraT forms a pore or if it simply forms an oligomeric structure on the cell surface. The authors should include an additional figure to illustrate this point.

Line 128: This statement references T137 and Figure 2D but T137 is not shown in Figure 2D.

Line 142: The authors note that the internal diameter of the pore is 39 Å in the text but indicate it is 48 Å in Figure 2B. This should be corrected.

Line 143: It is not clear what is meant by the “entry is mostly aligned by negatively charged residues”. This should be clarified.

Line 156: The oligomeric structure should be referred to as decameric.

Line 182: The link to the phylogenetic tree should be included in the data availability section.

Line 236: The authors discuss structural similarity of TraT homologs. It is assumed that the authors are comparing the structures determined in this manuscript to AlphaFold predictions. This should be made clear.

Line 241: The authors indicate that helices 3 and 4 are likely to adopt similar structures in TraT homologs despite poor predictions in AlphaFold. There should be some justification as to why this is expected.

Lines 244-245 are empty and can be deleted.

Line 277: Grammatical error: “it points on the role” should be changed to “it points to the role”.

SFX Assay Methods: The strains used in the conjugation assay are unclear and it is not immediately obvious how the transconjugants are being selected. The authors should clearly describe the assay in the methods section.

Figure 2: The raw data used to generate these plots should be included in the manuscript to allow the reader to appreciate the biological result.

Figure 4: The SEC profile depicted in panel A is somewhat complex compared to the SEC shown in panel B as there are ~3 additional peaks. Do these peaks also correspond to TraT? If so, it may indicate that the decameric species, although preferred, is not the only oligomer that forms spontaneously during purification.

Supplemental Figure 1: The meaning of the masses indicated for TraTpKpQIL and TraTF are not immediately obvious. The authors should include the expected masses of each protein alongside the corresponding spectra. Similarly, it is not clear whether the mass shift is the same for the two proteins nor is it obvious if it is expected to be. The authors should further clarify this point in the text to avoid confusion.

Supplemental Figures 2 and 4: The authors should include local resolution estimates.

Supplemental Figures 2 and 4: The authors should include model-map correlation statistics to indicate the quality of the models constructed. This is an important point as the resolution of the map fluctuates.

Supplemental Figure 3: The rotary symbol indicated seems to be incorrect.

File upload: 522738_0_related_ms_9297031_sgclhj_convrt.pdf does not contain any useful information though it does not appear to be required.

Version 1:

Reviewer comments:

Reviewer #1

(Remarks to the Author)

The authors have carefully addressed all of my prior concerns in this revised manuscript. The work nicely documents the novel structure of the TraT surface lipoprotein and significantly broadens our knowledge of how widely this protein is distributed among different bacterial phyla. Although the focus is on understanding how TraT might function in excluding redundant uptake of F plasmids among plasmid-carrying populations, the authors also discuss the importance of TraT in a medical context with demonstrated functions including immune evasion, inhibition of the complement cascade, and interaction with other host proteins. Due to its surface exposure, TraT also has been used to display foreign antigens on bacterial cell surfaces for therapeutic and other applications. Thus, definition of TraT's decameric structure extending from the cell surface and its phylogenetic distribution among bacteria is critical to a broader understanding of how this protein exerts its various functions. The clear and engaging style of writing and the novelty of the findings will attract a broad readership among scientists with interests in bacterial pathogenesis, cell-cell communication, and horizontal gene transfer. One correction to be made, the reference list doesn't have the correct citation for reference 25.

Reviewer #2

(Remarks to the Author)

The authors have revised the manuscript in detail. I have only one concern about the phylogenetic tree. The protein (ID: P0DKR9) produced an unusually long branch, the bootstrap supporting value at that position is also very low. I don't see a clear reason why this sequence must be included.

Reviewer #3

(Remarks to the Author)

The authors have adequately all addressed all comments I raised during the reievew.

We would like to thank all the Reviewers for their very positive and constructive criticism of our work. We have addressed all their queries in a point-by-point reply below:

Reviewer #1

Major:

1. Numbering of TraTpKpQIL protein sequence. I spent some time trying to figure out the numbering scheme the authors used for TraTpKpQIL based on the published sequence of plasmid pKpQIL (NC_014016.1). For some reason, the authors accessed a sequence (ARQ19738.1) that most likely incorrectly starts with a Met codon that is in frame with the start codon used for all other TraT homologs but resides 15 codons upstream. This yields an unusually long signal sequence of 35 residues, but more problematically a pre-protein of 258 residues as opposed to 243 residues in length - the lengths of other TraT homologs such as F that is also analyzed in this study. Upon processing at the conserved Cys residue, these TraT homologs are both 243 residues. Yet, by adopting the upstream Met as the (incorrect) start codon, TraTpKpQIL's residue numbers don't match those of TraT and most other TraT homologs, which complicates direct comparisons. I suggest two solutions. First, start TraTpKpQIL numbering at the downstream Met codon, so the pre-protein has 243 residues like other homologs.

For our functional and structural work we used the sequence that was originally isolated in Israel carrying KPC-3. The accession ID we used is KY798507.1 which is for pKpQIL-UK. This sequence is the one that carries the additional residues. To provide clarity, we have provided a sequence alignment between the TraT_{pKpQIL} and TraT_F that should make comparisons easier (Supplementary Figure 5a).

Second, include an alignment of TraTpKpQIL and TraTF, as well as an alignment of these proteins with a selection of other plasmid- and chromosomally-encoded TraT's. The value of these alignments is that they identify regions of variation among the homologs; having done this myself, it is evident that they cluster in only a few locations that can then be readily identified in the solved or AlphaFold-predicted TraT structures – these locations might mark functionally important sites that confer system specificity.

We added an alignment of the two sequences in Supplementary Figure 5a. In addition, we have also included a conservation hotspot map based on the plasmid and chromosomal TraTs that further highlight the regions that might be involved in specificity, Figure 6b. Reviewer #3 requested a similar figure.

2. L. 144. This truncation is testing for the combined importance of the DAG modification of C36 and alpha helix 1, not just the DAG modification, on TraT oligomerization in vitro. Although the tested mutation abolishes oligomerization, it cannot be concluded whether this is due to the loss of DAG modification, alpha helix 1, or both. To this end, a i) simple substitution of C to S and ii) a deletion of alpha helix 1 but with retention of C36 also should be made and tested for effects on TraT oligomerization in vitro. Additionally, with these mutations in hand, the authors should evaluate their in vivo effects on TraT SFX activity in matings by expression of the mutant proteins in recipient cells. This would allow for firm conclusions concerning the functional importance of lipid modification (OM anchoring) and the alpha helix 1 on protein activity in vivo.

In the original manuscript we wanted to test the effect of removing both DAG and TM helix a1 on the TraT. We agree that more subtle changes can point to their precise role. The C36S mutant resulted in no protein expression suggesting an important role of the lipidation of TM helix a1 and to process the mature TraT correctly. Retaining the C36 but removal of the TM helix a1 resulted in protein expression in the OM that was not able to confer SFX. Our attempt to purify the protein resulted in very low yields that were prone to aggregation (these data have not been included). These data are presented in Fig 4a and b and included in the main text.

3. Discussion. This could be strengthened considerably by:

i) Rewriting the first paragraph, which is pretty weak. Suggest highlighting the broader functional importance of TraT in SFX, serum resistance, resistance to lytic action of complement, as well as its early use for surface display of antigens of interest. There were also early reports that traT genes are widespread on F plasmids as well as other virulence plasmids independently of association with tra genes - see Micro Rev. 54:331.

We agree with the reviewer's point and we have re-written this section.

ii) Reducing the extent to which the Results are simply restated, and expanding on the broader biological significance of the findings.

We have amended the text accordingly.

iii) Last paragraph - Although the parallel roles of TraT and EspZ in blocking redundant transfer events are interesting, the mechanisms may be quite distinct – and the biological consequences clearly are (one blocking redundant transfer of a plasmid among bacteria, the other preventing transfer of too many effectors to a mammalian cell). As a last paragraph, however, the comparison distracts from the novelty and main findings of the present work, especially since it ends with the unfortunate conclusion that we still have no understanding of underlying mechanism in either instance. Suggest ending the discussion with a stronger paragraph that highlights the novelty/significance of the present work in the context of TraT's role in plasmid biology and ways this surface protein may contribute more broadly to infection or survival in the environment.

We agree with the reviewer point and we have re-written this section.

Minor:

4. Fig. 1b. Although it is not mentioned in the legend, I assume that the line represents the overall average of the data set being shown. If so, the line representing the transfer frequency into a recipient carrying pBAD-TraTF is aberrantly high, being in the middle of the two higher data points apparently without factoring in the lower data point.

The fitting of the data by Prism has taken into account the lower data point. We do not think there is bias towards the two higher data points.

5. L. 156. Fig. 5D. This is why the numbering should be made uniform – the numbering for residues in fig. 5D refers to TraTpKpQIL, forcing the reader to subtract 15 residues to identify the corresponding residues in TraTF.

See reply in point 1 please.

6. L. 169. Fig. 5. Panel C plus a sequence alignment would more clearly depict the variation between the two TraT's; by just showing Panel C, one cannot see which variations are conservative and which affect charge, hydrophobicity or size that might confer system specificity. The superimposition overrepresents the differences, which for the most part are highly conservative, e.g., G-S, S-A, T-A, V-A, K-R. The few differences of possible functional importance, e.g., at residues 93 (T-K), 105/107 (T-K, G-Y), 240/243 (P-E, Q-K), could be highlighted and, ideally, tested through mutational analyses (although this is not essential for the present manuscript).

Regarding the superimposition in Fig 5c, we are only highlighting that overall the two structures are near identical, and any selectivity is coming from subtle amino acid differences highlighted in panel 5d.

7. L. 196. The mature forms of TraTF and TraTR100 are 100 % identical – the single amino acid difference is in the signal sequence.

The reported value refers to the total number of sequences similar to TraTF and TraTR100.

8. L. 240. Direct the reader to Fig. 6C for the AlphaFold3 structures referred to in this sentence.

Added

9. L. 267. This would be more clearly presented with an alignment that highlights the few noteworthy differences in charge, hydrophobicity or charge (also, I count only 25 differences but I might be wrong).

In addition to an alignment, Supplementary Figure 5, we have also included a conservation hotspot map based on the plasmid and chromosomal TraTs that further highlight the regions that might be involved in specificity, Figure 6b. Reviewer #3 requested a similar figure.

Figure 5d does contain 27 differences.

10. L. 287. Add the citation here.

Added

11. Supplementary Fig. 4 legend – appears to be copy-pasted from Suppl. Fig. 2 - Change references of "TraTpKpQIL" to "TraTF."

Corrected

12. Supplementary Table 2. GFP-D – it's mentioned that pKpQIL and sfGFP controlled by Plac. What does this mean? Assuming that sfGFP expression is controlled by Plac, does Plac control something else on pKPQIL, e.g., transfer? Explain. Also, for rigor/reproducibility, the source should be a prior publication, not an author's name. This is also valid for pBAD-traTpKpQIL – was this constructed here or was it published previously, those are the two options, not an author's name as the source. Finally, the more conventional designation is Ery, not Ert, assuming this is the abbreviation for erythromycin.

We have clarified the methods section. The reporter pKpGFP (which is now defined in the methods) expresses sfGFP . We have added the reference to prior work.

Reviewer #2

Since I am not an expert in solving protein structures, below are my concerns and questions mainly regarding the genomic analysis.

Line 26-28, 86-87: The phylogenetic analysis is disconnected from the former results, lacking a clear motivation.

We believe that our structural and phylogenetic insights of TraT provide complementary insights that will together pave the way for further elucidating the mechanism of TraT SFX. For example, the identification of diverse homologues provides a basis for further exploring the structural impact of sequence variations, building on the novel insights of the TraT structure elucidated here. In particular, vital clues relating to the SFX mechanism may lie in the comparison of TraT from plasmids in Enterobacteriaceae, Legionellaceae and the individual plasmids from *P. salmonis*, *S. kujjense* and *A. aromaticum*

Line 28-29: this sentence interrupts the structure. The author should first hypothesize what was expected. For example, TarT on chromosomes has never been reported.

We have include a sentence to describe the novelty of chromosomal TraTs.

Line 182: Before phylogeny reconstruction, please control the alignment quality. Sequences producing low alignment quality can be observed in Figure 6a, corresponding to unusually long branches. Please add bootstrap support and indicate the protein evolution model and rooting method. The author should consider cutting the tree by above 65% identity to TraTpKpQIL, long branch effects have a detrimental impact on phylogeny reconstruction and rooting. When the conclusion is the independent evolution of TarT in Enterobacteriaceae and Legionella, Figure 6a should mainly display this result.

We thank the reviewer for these excellent suggestions. We have now used the tool ModelTest-NG (<https://academic.oup.com/mbe/article/37/1/291/5552155>) to select the best-fitting model for the protein alignment (LG+G4), as described in the text. We then constructed a maximum-likelihood tree using RAXML-NG using this model with 1000 bootstrap replicates. We have generated a new figure, Figure 6, which shows this tree, with bootstrap values also available to view via Microreact (<https://microreact.org/project/tra-t>). There have been some minor alterations to the topology from the previous version of the

tree although the main conclusions drawn remain unchanged. The tree is midpoint-rooted as we have no known outgroup, as now more clearly indicated in the text and legend of Figure 6. We have kept all the original protein sequences in the tree (with $\geq 30\%$ identity to TraT(pKpQIL) as using a cut-off of 65% identity would result in only TraT from Enterobacteriaceae IncF plasmids being retained, and we would thus lose the conclusions regarding the independent evolution of TraT in Enterobacteriaceae and Legionellaceae plasmids, and the high number of divergent lineages containing chromosomally-encoded TraT. We have also reviewed our language used to describe the topology of the tree to account for the rooting being unknown and the subsequent uncertainties relating to the root/origin of TraT. The new tree figure (Figure 6) also now clearly shows the two separate groupings of TraT in Enterobacteriaceae and Legionella.

Line 195: “Some clustering” is imprecise. I suggest reporting the percentage of TraT variants found in each sub-lineage (clade) on the phylogenetic tree. The author should perform constrain tree topology tests on clustering/ lineage.

The genus from which each TraT sequence is derived is now shown in Figure 6, allowing a visual inspection of the clustering. We have now also provided more precise details/examples in the text:

“For example, TraT_F and TraT_{R100} (from *E. coli*) belonged to a clade in which the majority (82.6%; 100/121) of sequences were derived from either *Escherichia*, *Shigella* or *Salmonella*, while TraT_{pKpQIL} (from *K. pneumoniae*) was located in a clade in which the majority of sequences (83.3%; 50/60) were from the *Klebsiella/Raoultella* genus.”

We have also added some specific examples regarding the clustering of TraT variants by IncF subtype:

“We also found that TraT sequences from plasmids carrying the same replicons usually shared high similarity, such as those with IncFIB(S)/IncFII(S) (97.1-100% identity among 13 plasmids). However, a higher diversity of TraT sequences was found among some replicon types including IncFII(pCoo) (77.0-100% among 12 plasmids) which were found in different sub-clades of the plasmid-associated clade.”

Tree topology tests are used to compare alternative phylogenies so are not appropriate here where we are simply stating that there are associations between certain traits (e.g. host genus, IncF subtype) and particular clades of the tree. However, the bootstrap values for all individual clades are available to view via Microreact, as now mentioned in the text.

Line 199-202: The reported replicon info (Col(pHAD28)) here is not precise, PlasmidFinder identifies marker sequence without considering the plasmid backbone. pSTN0717-36-2 (accession: AP022496) is a 118,786 bp plasmid and encodes a F-like repA1 gene (locus tag: STN0717CIT36_P20010). This plasmid likely results from a plasmid fusion event (a small size ColE1-like plasmid and a large size F-like plasmid), similar to the multiple-replicon plasmids reported in the interactive tree display (Col156, IncFIA, IncFIB, IncFII; ColRNAI). The repB replicon is not a standard replicon type.

We thank the reviewer for this comment. We have now inspected the *tra* operons of these “non-IncF” plasmids and found that they do indeed all carry an F-like *tra* operon. We have thus amended the text:

“We also identified seven TraT homologues in this clade possessing replicon types other than IncF subtypes including Col(pHAD28), IncR and repB. However, all seven plasmids also possessed an F-like *tra* operon with *traT* in the same position as in IncF plasmids, suggesting that they may represent fusion events of different plasmid backbones.”

Line 203; 220; 228: Expressions like a clade/ a sub-clade/ this clade, are not very reader-friendly, readers have to figure these clades out on the interactive tree display, with only figure 6a, it's impossible.

We have now amended the figure (Figure 6) which clearly marks the clades comprised of TraT from the Enterobacteriaceae IncF/F-like and Legionellaceae F-like plasmids.

Line 209-212, 228-235: This part of comparative analysis is of special interest. It seems like *traT* is mobile via unknown means (or assembly artifacts). This should be discussed based on observations from the flanking sequences.

We agree that the mobility of *traT* is very interesting. Analysis of the flanking regions hasn't yielded any obvious means of mobility. However, this is perhaps unsurprising as the number of observed mobilisation events observed is quite small (i.e. many more might be expected if there was still an obvious intact mechanism). We are also confident that these are not assembly artefacts as chromosomally-encoded TraT sequences from the plasmid-associated Enterobacteriaceae clade often cluster together, which would be unexpected if these were due to random assembly-based issues.

Line 211-212: I manually checked the functional annotation of the flanking protein-coding genes in plasmid pPC1000_1 (accession CP059403), there is a *tar*-like operon from *tarD* (locus tag: FOLKNPGA_03766) to *traT* (locus tag: FOLKNPGA_03787). The protein sequences may be quite different from the other five *tar* operons in Figure S5.

We thank the reviewer for this comment. Indeed, our initial observation was incorrect and there is a *tra*-like operon in the pPC1000_1 plasmid, albeit with the protein sequences highly diverged from those in other *tra* operons of Legionellaceae plasmids. We have thus removed the sentence stating that *traT* could not be found within an intact *tra* operon within the pPC1000_1 plasmid. For simplicity, we have also removed this plasmid from the associated supplementary figure (Figure S6).

Line 223-227: I found the expression “additional” here confusing. To me, “additional” refers to traT duplications (in chromosomes and plasmids).

For clarity, we have amended this sentence to:

“In addition to the plasmid-associated clades described above, three other plasmid-encoded TraT homologues from *Piscirickettsia salmonis*, *Sulfuricurvum kujiense* and *Aromatoleum aromaticum* were found in separate lineages of the tree with each sequence most closely-related to chromosomal TraT sequences from other species.”

Line 231-232: Please provide the sequence location/reference for this genomic island.

We have now added the accession number for this chromosomal sequence and the coordinates for the genomic island.

Line 236: The high structural similarity is AF3-predicted. This should be clarified.

We modified the text to make it clearer.

Line 283-284: If a phylogenetic root is identified in the chromosomal clade on the tree, the suggestion here could be more trustworthy. (Note: this should be tested)

As discussed above, we have midpoint-rooted the phylogenetic tree as there is no known outgroup. We cannot therefore know whether the true root of the tree lies in the chromosomal clade. Nevertheless, the high diversity of chromosomal TraT sequences and their positioning among many numerous divergent lineages suggests that chromosomal TraT sequences have ancient origins. However, based on our tree, we agree that it is not possible to say whether chromosomal or plasmid TraTs came first and have thus amended the sentence.

“Taken together, these findings suggest that *traT* has early origins as a chromosomal gene, despite its more familiar role in SFX in the *Enterobacteriaceae* IncF plasmid *tra* operon.”

Line 65: refractory -> refractory

We have amended this.

Reviewer #3

Major Considerations

1. The negatively charged ring along the outer rim of the extracellular face of TraTpKpQIL is interesting though it is not clear if this feature is conserved in TraTF. The authors should generate an electrostatic map of TraF for comparison. Similarly, residues within this feature

should be mutated and the SFX efficiency evaluated using the conjugation assays described.

We added an electrostatic map for TraTF, Supplementary Fig 3. A very similar charge distribution is also observed as the residues are conserved. There is an additional positive charge from T92K in TraTF (see Fig 5d) but the overall profile is very similar.

Reviewer #1 also commented on possible mutagenesis (see point 6) but we believe that specificity is not driven by a single amino acid and it would require substantial work of mutant combinations to pin point the exact nature of specificity. The Reviewer also commented that this is not necessary in the context of this work.

The authors note that a TraTpKpQIL mutant missing helix 1 is no longer able to oligomerize in vitro and attribute this function to DAG binding. The authors should mutate residues within the DAG binding site and the modified cysteine residue and perform similar SEC experiments to illustrate the role of DAG in oligomerization.

Reviewer #1 raised a similar question. Instead of mutating the interacting residues, we mutated C36S and we observe total loss of protein expression. The data have been added in the text and in Fig 4a and b. Loss of the $\alpha 1$ but retaining the DAG site resulted to some protein in the OM but incapable of providing SFX. Our attempt to purify the protein resulted in very low yields that were prone to aggregation (these data have not been included).

A list of substitutions are plotted on the TraTpKpQIL structure in Figure 5D which appear to be scattered about the structure. However, because TraT is forming oligomeric structures, it is possible that these substitutions map to three-dimensional sites on the decamer and not the monomer. The authors should generate a figure showing these substitutions on the decamer structure.

Most of the substitutions are found on the surface of the β -sandwich domain. We have generated the requested figure, Supplementary Fig 5.

The authors have assembled 288 TraT sequences and found some divergent sequences. The conservations of residues found in these sequences should be mapped to the structures of TraT reported here to illustrate variability across species and genetic locus (plasmid vs. chromosome).

A very good point. We have mapped them onto the structure using ConSurf and present them in Supplementary Fig 5.

The formation of the decamer structure is intriguing but no functional rationale is presented as to why TraT must form these oligomeric structures to function. Similarly, it is not clear if the monomer is incapable of performing SFX. The authors should evaluate the efficiency of TraTpKpQIL monomeric mutants in conjugation assays.

This is a very interesting point. To break the decamer it will require extensive mutagenesis and it may result in folding issues. Judging from the soluble monomeric version that is prone to aggregation and OM associated mutant, with a1 absent, that is not capable to confer SFX, we do not believe that a monomer is functional. The nature of the oligomer could be also to its other functions with complement pathway inhibition.

Minor Suggestions

Line 49: Grammatical error – “we have recently showed” should be “we have recently shown”

Corrected

Line 103: The oligomeric structure should be referred to as decameric.

Changed

Line 106: The authors refer to the profile of TraTpKpQIL over a SEC column but do not refer to the data which is presumably included in Figure 4. The authors should reference this result.

We deleted the sentence as Fig 4 would be referenced earlier and it would not conform with Figure calling. Despite this, the profile is shown as part of the oligomeric discussion section.

Line 116: Grammatical error: “displays a 10-fold symmetry” should be changed to “displays 10-fold symmetry”.

Corrected

Line 119: The authors indicate that a belt of featureless density surrounds the helical segment of TraTpKpQIL but do not indicate if additional density is found within the helical belt. Without clarification it is not clear if TraT forms a pore or if it simply forms an oligomeric structure on the cell surface. The authors should include an additional figure to illustrate this point.

We amended the text to make it clear that density is also found within the barrel domain. The right panel of Fig 2a shows the additional density in the barrel.

Line 128: This statement references T137 and Figure 2D but T137 is not shown in Figure 2D.

We moved the reference earlier in the sentence as we prefer to omit T137 for clarity.

Line 142: The authors note that the internal diameter of the pore is 39 Å in the text but indicate it is 48 Å in Figure 2B. This should be corrected.

Corrected

Line 143: It is not clear what is meant by the “entry is mostly aligned by negatively charged residues”. This should be clarified.

We referred to the top part of the opening. We have changed the text to make it clearer.

Line 156: The oligomeric structure should be referred to as decameric.

Changed

Line 182: The link to the phylogenetic tree should be included in the data availability section.

Done

Line 236: The authors discuss structural similarity of TraT homologs. It is assumed that the authors are comparing the structures determined in this manuscript to AlphaFold predictions. This should be made clear.

Yes this is correct. The text has been amended.

Line 241: The authors indicate that helices 3 and 4 are likely to adopt similar structures in TraT homologs despite poor predictions in AlphaFold. There should be some justification as to why this is expected.

This is based on our cryo-EM structure and AF3 model for TraTpKpQIL/F. We have amended the text to make it clearer.

Lines 244-245 are empty and can be deleted.

done

Line 277: Grammatical error: “it points on the role” should be changed to “it points to the role”.

Corrected

SFX Assay Methods: The strains used in the conjugation assay are unclear and it is not immediately obvious how the transconjugants are being selected. The authors should clearly

describe the assay in the methods section.

We have amended the materials and methods to make it clearer.

Figure 2: The raw data used to generate these plots should be included in the manuscript to allow the reader to appreciate the biological result.

We assume the reader is referring to Fig 1 as Fig 2 contains the cryo-EM maps that will be available through the PDB and EMDB. The raw SFX data will be submitted to Nature Communications as per their publication policy.

Figure 4: The SEC profile depicted in panel A is somewhat complex compared to the SEC shown in panel B as there are ~3 additional peaks. Do these peaks also correspond to TraT? If so, it may indicate that the decameric species, although preferred, is not the only oligomer that forms spontaneously during purification.

This a very good point. We see a very similar profile for TraTF. In our class averages we only see decamers and the broader peak is probably due to the shape of TraT and resolvability of the SEC. The 8ml peak is close to the void volume so very likely some aggregation overlapping with the main peak.

Supplemental Figure 1: The meaning of the masses indicated for TraTpKpQIL and TraTF are not immediately obvious. The authors should include the expected masses of each protein alongside the corresponding spectra. Similarly, it is not clear whether the mass shift is the same for the two proteins nor is it obvious if it is expected to be. The authors should further clarify this point in the text to avoid confusion.

We amended the figure legend to make it clearer how the masses add up. It is unclear from the literature how the DAG acyl chain is selected (DAG ranges between 300-800 Da). In our mass spectrometry data we see a difference of 200 Da between the TraTpKpQIL and TraTF suggesting difference in the processing.

Supplemental Figures 2 and 4: The authors should include local resolution estimates.

The estimates have been included.

Supplemental Figures 2 and 4: The authors should include model-map correlation statistics to indicate the quality of the models constructed. This is an important point as the resolution of the map fluctuates.

The requested figures have been included from the validation reports.

Supplemental Figure 3: The rotary symbol indicated seems to be incorrect.

The figure has been changed from the original submission and we have not included a symbol in the revised figure.

Reviewer #2

The authors have revised the manuscript in detail. I have only one concern about the phylogenetic tree. The protein (ID: P0DKR9) produced an unusually long branch, the bootstrap supporting value at that position is also very low. I don't see a clear reason why this sequence must be included.

The protein has been excluded from the tree and Figure 6 has been updated.